# Decision Information Meets Large Language Models: The Future of Explainable Operations Research

**Yansen Zhang**[1*]**, Qingcan Kang**[2†]**, Wing Yin Yu**[2]**, Hailei Gong**[2]**, Xiaojin Fu**[2]**, Xiongwei Han**[2]**, Tao Zhong**[2]**, Chen Ma**[1†]
[1]Department of Computer Science, City University of Hong Kong
[2]Huawei Noah's Ark Lab
`yanszhang7-c@my.cityu.edu.hk`
`kangqingcan@huawei.com`
`chenma@cityu.edu.hk`

## Abstract

Operations Research (OR) is vital for decision-making in many industries. While recent OR methods have seen significant improvements in automation and efficiency through integrating Large Language Models (LLMs), they still struggle to produce meaningful explanations. This lack of clarity raises concerns about transparency and trustworthiness in OR applications. To address these challenges, we propose a comprehensive framework, Explainable Operations Research (EOR), emphasizing actionable and understandable explanations accompanying optimization. The core of EOR is the concept of *Decision Information*, which emerges from what-if analysis and focuses on evaluating the impact of complex constraints (or parameters) changes on decision-making. Specifically, we utilize bipartite graphs to quantify the changes in the OR model and adopt LLMs to improve the explanation capabilities. Additionally, we introduce the first industrial benchmark to rigorously evaluate the effectiveness of explanations and analyses in OR, establishing a new standard for transparency and clarity in the field.

## 1 Introduction

Operations Research (OR) has a long history of optimizing complex decision-making processes, such as in logistics, finance, investment, transportation, and healthcare, etc., where even small improvements can lead to significant operational profits. As these optimization algorithms increasingly contribute to daily life, it is essential to ensure their trustworthiness and reliability through explanations, which build user confidence (Faulhaber et al., 2021). Governments are also responding to this need by enacting laws like the General Data Protection Regulation (GDPR) of the European Union (Goodman & Flaxman, 2017), emphasize the "right to explanation" for algorithmic decisions (Selbst & Powles, 2018) in automated systems.

In recent years, Large Language Models (LLMs) have emerged as powerful tools in the OR domain, offering new opportunities to automate and enhance the modeling process. Current research of LLMs in OR, such as works (Xiao et al., 2023; AhmadiTeshnizi et al., 2024; Tang et al., 2024; Zhang et al., 2024), explore the potential to streamline the formulation and solutions of complex OR problems. However, LLMs in OR have primarily focused on improving efficiency and accuracy by generating codes for external solvers to obtain OR solutions, with less attention to enhancing solution explainability, especially in real-time collaborative automated systems.

Meanwhile, several studies (Čyras et al., 2019; Li et al., 2023; Erwig & Kumar, 2024; De Bock et al., 2024) have explored explainable optimization related to OR, but there are still limitations. For example, (Erwig & Kumar, 2024) focuses specifically on combinatorial optimization problems,

---

[*] Work done as an intern in Huawei Noah's Ark Lab.
[†] Corresponding authors.

which limits its applicability across the broader OR landscape and fails to leverage the advantages of LLMs in real-time modeling and explanations. Another work, OptiGuide (Li et al., 2023), emphasizes what-if analysis, which, while useful for specific easy scenarios, lacks the robust modeling capability to address more complex cases like deleting or combining constraints. For example, if a warehouse closes, the OR model must remove the related storage capacity constraint and adjust the distribution network. Current methods struggle to achieve this level of flexibility, yet such adaptability is crucial for accurately reflecting real-world changes. Most critically, the explanations these methods provide are often superficial, merely summarizing the outcomes without exploring the underlying reasons behind the results, thus lacking the quantitative analysis, depth, and clarity required to fully understand and trust the decision-making process.

Given the limitations of existing approaches to explainable optimization, we are motivated to develop a more comprehensive framework, EOR, for explaining OR models. As shown in Figure 1, our framework addresses the critical need for transparency in OR by shifting from purely modeling a problem (a natural language description) to providing clear, actionable explanations for a user query, such as *"What if transportation costs increase by 15%?"*. First, we formulate the problem of explainable OR within the context of LLMs. This formulation is essential for laying a foundation for future research in this emerging area. Second, our framework emphasizes two critical types of explanations, 1) *Explanation of correctness:* the reasons for code updates during the modeling process, and 2) *Explanation of the Results:* the rationale for generating specific solutions. Unlike traditional methods that provide superficial explanations and analyses, our approach incorporates more sophisticated what-if analysis, quantifying the effects of changes prompted by user queries and providing deeper insights into the decision-making process. A significant component of our contribution is the introduction of the concept of "Decision Information". By leveraging bipartite graphs, we measure the importance of different decision factors in response to user queries, particularly when complex changes in constraints arise. This approach enhances both the modeling capabilities and the explanatory power of OR models within LLMs. Our dual focus ensures that the framework not only yields accurate optimal solutions but also effectively communicates the underlying rationale for these solutions. Finally, recognizing the need for standardized evaluation of explainable OR methods, we design a new industrial benchmark specifically tailored to assess the effectiveness of explanations in OR. This benchmark fills the gap of current approaches and sets a new standard for evaluating the transparency and comprehensibility of OR models.

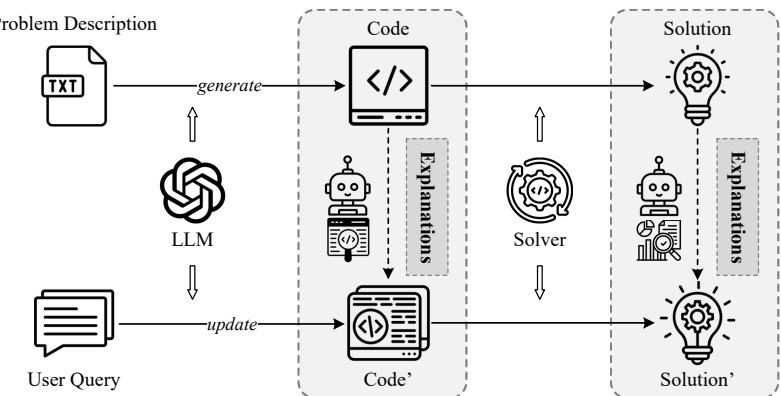

Figure 1: The framework of EOR.

**Contributions.** 1) We formulate the problem of the explainable OR problem within the context of LLMs, laying a foundation for future research in this area. 2) We introduce the concept of "Decision Information" and utilize bipartite graphs in conjunction with LLMs to quantify its importance in response to user queries, enhancing both the modeling capabilities and the explanation of complex what-if analysis within OR. 3) We develop a new benchmark specifically designed to evaluate the effectiveness of explanations in OR, setting a new standard for explainability in the field.

## 2 RELATED WORK

### 2.1 LLMs FOR OR

LLMs show great promise for OR, offering innovative approaches to optimize and automate modeling processes (Xiao et al., 2023; AhmadiTeshnizi et al., 2024; Tang et al., 2024; Zhang et al., 2024; Huang et al., 2024; Mostajabdaveh et al., 2024). Although LLMs have shown potential in various OR tasks, their application has primarily focused on automating modeling processes and enhancing computational efficiency. In contrast, our approach distinguishes itself by using LLMs to provide detailed, context-aware explanations of OR solutions, addressing the gap in explainability.

### 2.2 EXPLANATIONS FOR OR

Explanation in OR is essential for clarity and transparency, helping various stakeholders understand complex decision-making processes. Despite its significance, there is a notable lack of research, with only a few related works, such as (Thuy & Benoit, 2024; Erwig & Kumar, 2024; De Bock et al., 2024) and OptiGuide (Li et al., 2023), which only focus on the easy what-if analysis. Our approach focuses on more complex what-if analysis, seeking a broad range of methods that can uniformly quantify constraint changes or parameter changes (sensitivity analysis). We leverage LLMs to embed detailed, context-aware explanations directly within OR solutions, filling this gap. The comparison between EOR and OptiGuide is shown in Table 1. More details about the differences between the concepts are discussed in Appendix A.1.

Table 1: The comparison of OptiGuide and EOR.

| | What-if Analysis | | Decision Information Analysis | Sensitivity Analysis |
|---|---|---|---|---|
| | Easy | Complex | | |
| OptiGuide | ✓ | ✗ | ✗ | ✗ |
| EOR | ✓ | ✓ | ✓ | ✓ |

## 3 METHODOLOGY

### 3.1 PROBLEM FORMULATION

Given an OR problem $\mathring{p}$, along with a user query $\mathring{q}$ related to the problem, our goal is to utilize an LLM to generate comprehensive explanations of solutions for the queries in real time. The LLM will provide two types of explanations: 1) *Attribution Explanation*, which outlines the general attributes and structure of the problem, and 2) *Justification Explanation*, which elucidates the correctness and derivation of the solutions. The mathematical formulation is as follows:

**Input:** The origin problem description $\mathring{p} = \langle \mathring{d}, \mathring{o}, \mathring{c} \rangle$, and a user query $\mathring{q} = \langle \mathring{o}', \mathring{c}' \rangle$. The updated problem description, incorporating the user query, is denoted as $\mathring{p}' = \langle \mathring{d}, \mathring{o}', \mathring{c}' \rangle$. Here, $\mathring{d}$ represents the set of decision variables, $\mathring{o}$ and $\mathring{o}'$ are the objective functions to be maximized or minimized, $\mathring{c}$ and $\mathring{c}'$ denote the original and modified constraints in $\mathring{p}$ and $\mathring{p}'$, respectively, that the decision variables must satisfy. In our setting, we assume the decision variables remain unchanged.

**Output:** We denote the problem solutions for $\mathring{p}$ and $\mathring{p}'$ as $\mathring{s}$ and $\mathring{s}'$, respectively. The output comprises two types of explanations, 1) *Attribution Explanation*: A detailed description of the elements $\mathring{d}$, $\mathring{o}$, $\mathring{o}'$, $\mathring{c}$, $\mathring{c}'$, $\mathring{s}$, and $\mathring{s}'$ within the context of the problem. 2) *Justification Explanation*: A rationale for the correctness of $\mathring{s}$ and $\mathring{s}'$, and clarifies how $\mathring{s}'$ is derived from $\mathring{s}$.

### 3.2 THE EOR FRAMEWORK

Our proposed framework, EOR, is an end-to-end solution designed to enhance OR model transparency using LLMs. Unlike current methods that provide limited and shallow explanations, primarily the form of attribution explanations, our framework emphasizes delivering clear, actionable

insights for diverse stakeholders. As illustrated in Figure 1, we focus on the justification explanations. We will offer two critical explanations: 1) justifications for code updates during the modeling process, and 2) the rationale behind specific solutions. By doing so, we strive to make OR solutions more accessible, understandable, and applicable to a broader audience, thereby improving decision-making quality and fostering user trust.

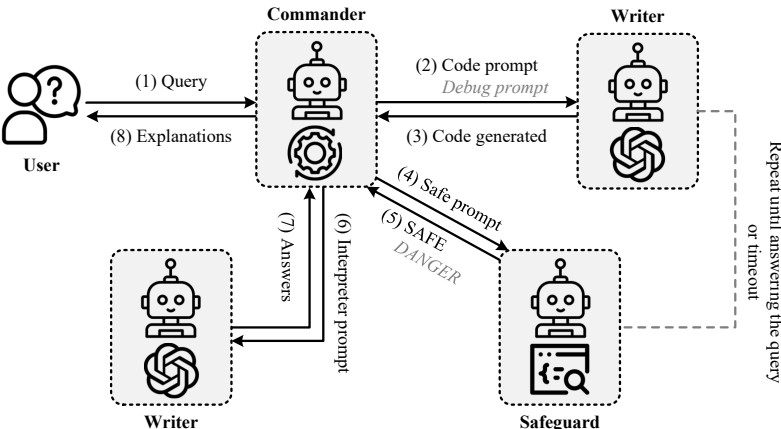

Figure 2: The overall workflow of EOR.

### 3.2.1 THE WORKFLOW OF EOR

As shown in Figure 2, EOR framework comprises three key agents: Commander, Writer, and Safeguard, each serving a distinct role to ensure an efficient, accurate, and secure optimization process.

**Commander Agent:** The Commander acts as the central hub or "data bus" in the system, responsible for receiving user queries and managing data flow between the agents. When an end-user submits a new user query, the Commander first interprets the query's context, identifies the intent, and then forwards the message to the appropriate agents.

**Writer Agent:** Upon receiving the processed query from the Commander, the Writer initially assumes the role of analyzing and modifying the code. Based on the query's requirements, the Writer determines whether to add, delete, or update specific constraints and parameters. By leveraging LLMs, the Writer guarantees that the generated code accurately reflects the intended changes. Subsequently, the updated code is sent to the Safeguard for verification. Once the Safeguard provides a SAFE confirmation, the Writer transitions into an interpreter role, generating detailed explanations for the modifications and the rationale behind the decision-making process.

**Safeguard Agent:** The Safeguard is responsible for ensuring the safety and correctness of the generated code. It conducts thorough checks to verify whether the code adheres to predefined safety standards and is free from logical or syntactical errors that could compromise the optimization process. If the code passes these safety checks, the Safeguard approves it for execution; otherwise, it triggers a debugging process where the Writer regenerates and corrects the code as necessary.

**The overall workflow:** The EOR workflow starts when a user submits a query to the Commander (1), who relays it to the Writer with a code prompt (2). The Writer analyzes the query, determining whether to add, delete, or update code, and returns the updated code to the Commander (3). This step ensures that the generated code aligns with the updated problem requirements. The Commander then sends the code to the Safeguard for verification (4). If the Safeguard determines the code is safe (5), the process moves to (6), where the Commander sends an interpreter prompt to the Writer. If the code is deemed dangerous, the process loops back to (2), with the Commander sending a debug code prompt to the Writer. Once the code is safe, the Writer generates answers about explanations for the modifications and results (7) and sends them to the Commander. Finally, the Commander sends these explanations to the user (8). This iterative process, which repeats until a satisfactory answer or timeout, ensures robust, explainable solutions tailored to the user's query. The detailed design of the prompt template is presented in Appendix A.2.

### 3.2.2 JUSTIFICATION EXPLANATION GENERATION

In our framework, explanations are generated to ensure transparency and trustworthiness in the decision-making process. These explanations are divided into two main categories:

**Explanation of Correctness:** This type of explanation serves to validate the code modifications introduced by the Writer, offering a detailed rationale for the changes made. It clarifies the necessity of these modifications in addressing the problem's requirements and ensures that they adhere to safety standards and logical constraints. Through this process, the reliability of the generated code is substantiated, thereby enhancing the accuracy of the model prior to execution.

**Explanation of the Results:** Once the code is executed and the results are obtained, this type of explanation focuses on interpreting the outcome. It breaks down the results into understandable terms, illustrating the impact of the code changes on the final solution. This explanation connects the modifications to their direct effects, providing users with a clear understanding of how the new solution addresses their initial query and any resulting trade-offs or benefits.

Before formalizing the concept of "Decision Information", consider the following user query from an OR problem in flight operations: *How should the aircraft configuration be adjusted if the company limits Type A aircraft to 15 and Type B aircraft to 30?* In this context, "Decision Information" refers to the new constraints, limiting the number of Type A and Type B aircraft to 15 and 30, respectively, which directly modify the optimization model. These changes reshape the solution space, requiring adjustments to meet demand within the newly imposed limits. Thus, "Decision Information" captures the key query elements that drive changes in the problem.

**Definition 1** *Decision Information: Decision Information encompasses the parameters and constraints specified in a user's query for an OR problem, capturing the essential details needed to reconfigure the problem according to the user's intent.*

Following this definition, we turn to a quantitative evaluation of "Decision Information" to assess its impact on decision-making processes. However, existing methods lack a measure for assessing changes in decision information caused by constraint modifications, focusing only on sensitivity analysis in parameter changes. Inspired by (Xing et al., 2024), we convert both the updated and original programs into a standardized format and then calculate their differences to determine the importance of information changes. This process can be outlined in three steps:

**Conversion to Linear Programs (LPs) in General Form:** Both the updated code (resulting from user queries) and the original code are first parsed and translated into a standardized LP format, that is widely adopted by various LP solvers, including CPLEX (Cplex, 2009), Gurobi (Bixby, 2007), COPT (Ge et al., 2023) an so on. This conversion includes expressing all decision variables, objective functions, and constraints uniformly to allow a direct comparison. The general LP form captures the essence of both the initial and modified decision scenarios, providing a clear basis for analyzing how changes in input data or constraints affect the outcome.

Formally, an LP with $n$ variables and $m$ constraints can be represented as:

$$\min_{\boldsymbol{x} \in \mathbb{R}^n} \quad \boldsymbol{c}^\top \boldsymbol{x}$$
$$\text{s.t.} \quad \boldsymbol{l}^s \leq \boldsymbol{A}\boldsymbol{x} \leq \boldsymbol{u}^s$$
$$\boldsymbol{l}^x \leq \boldsymbol{x} \leq \boldsymbol{u}^x,$$

where $\boldsymbol{A} \in \mathbb{R}^{m \times n}$ is the constraint matrix, $\boldsymbol{c} \in \mathbb{R}^n$ is the cost vector, $\boldsymbol{x} \in \mathbb{R}^n$ is the decision variables, $\boldsymbol{l}^x \in \mathbb{R}^n$ and $\boldsymbol{u}^x \in \mathbb{R}^n$ are lower/upper bound of the decision variables $\boldsymbol{x}$, and $\boldsymbol{l}^s \in \mathbb{R}^n$ and $\boldsymbol{u}^s \in \mathbb{R}^n$ are lower/upper bound of the constraint.

**Graph Representation Conversion:** The standardized LPs can be transformed into bipartite graphs, which consist of two distinct sets of nodes: decision variables and constraints (Gasse et al., 2019; Fan et al., 2023; Xing et al., 2024). In this representation, edges between nodes signify dependencies or relationships, with edge weights indicating the strength or nature of these connections. We define the bipartite graph as $\mathcal{G} = (S \cup X, E)$, where $S = \{s_i | i \in [m]\}$ represents the set of constraint nodes, $X = \{x_j | j \in [n]\}$ represents the set of decision variable nodes, and $E = \{e_{i,j} | i \in [m], j \in [n]\}$ represents the edges between them. Here, $[\cdot]$ denotes a set of consecutive numbers. The attribute of a constraint vertex $s_i$ is expressed as $\text{attr}(s_i) = [l_i^s, u_i^s]\top$, indicating

its lower and upper bounds. Similarly, the attribute of a decision variable vertex $x_j$ is given by $\text{attr}(x_j) = [l_j^x, u_j^x, c_j]\top$, which includes its lower bound $l_j^x$, upper bound $u_j^x$, and objective coefficient $c_j$. This graph-based approach facilitates a structural analysis of the decision-making framework, allowing us to visualize the interactions among variables under different conditions and to compare updated and original programs on a structural level.

**Graph Edit Distance (GED) Calculation:** To quantify the impact of changes in "Decision Information", we compute the GED between the two bipartite graphs derived from the updated and original LPs. GED represents the minimum cost necessary to transform one graph into another through a sequence of operations, such as inserting, deleting, or substituting vertices and edges. This metric effectively captures the types of modifications made to the code in response to a query. This metric quantifies the minimal number of modifications (such as adding, deleting, or substituting nodes or edges) required to transform one graph into the other. A smaller edit distance indicates fewer changes, suggesting the updated program closely aligns with the original decision-making context. Conversely, a larger edit distance highlights significant alterations, signaling a substantial impact of the updated information on the decision-making process.

There are many well-established GED algorithms (Gao et al., 2010; Stauffer et al., 2017; Abu-Aisheh et al., 2015; Xing et al., 2024), we follow a straightforward principle provided by (Xing et al., 2024): each operation on an attribute in graph incurs a unit cost of 1. Formally, given the graph of the original program $\mathcal{G}^p = (S^p \cup X^p, E^p)$ and updated program $\mathcal{G}^{p'} = (S^{p'} \cup X^{p'}, E^{p'})$, we define the vertex cost matrix as follows,

|  | $s_{i'}^p \in S^p$ | $x_{i'}^p \in X^p$ | $\epsilon$ |
|---|---|---|---|
| $s_i^{p'} \in S^{p'}$ | $\#\text{msm}(s_i^{p'}, s_{i'}^p)$ | $\infty$ | $\#\text{attr}(s_i^{p'})$ |
| $x_i^{p'} \in X^{p'}$ | $\infty$ | $\#\text{msm}(x_i^{p'}, x_{i'}^p)$ | $\#\text{attr}(s_i^{p'})$ |
| $\epsilon$ | $\#\text{attr}(s_{i'}^p)$ | $\#\text{attr}(s_{i'}^p)$ | $\infty$ |

In this context, all operations are treated as matching processes; for example, deleting a vertex is conceptualized as matching the vertex to an empty vertex, denoted by $\epsilon$. Here, $\#\text{attr}(\cdot)$ denotes the total number of vertex attributes, while $\#\text{msm}(\cdot)$ represents the count of mismatched attributes between two vertices. Similarly, the edge cost matrix is defined as follows,

|  | $e_{i,j}^p \in E^p$ | $\epsilon$ |
|---|---|---|
| $e_{i,j}^{p'} \in E^{p'}$ | $\#\text{msm}(e_{i,j}^{p'}, e_{i,j}^p)$ | $\#\text{attr}(e_{i,j}^{p'})$ |
| $\epsilon$ | $\#\text{attr}(e_{i,j}^p)$ | $\infty$ |

Given the varying scales of changes in parameters and constraints, we further normalize $\text{GED}(\mathcal{G}^{p'}, \mathcal{G}^p)$ by the graph size. The normalized GED (NGED) is calculated as: $\text{NGED}(\mathcal{G}^{p'}, \mathcal{G}^p) = \frac{\text{GED}(\mathcal{G}^{p'}, \mathcal{G}^p)}{\max(|\mathcal{G}^{p'}|, |\mathcal{G}^p|)}$, where $|\mathcal{G}|$ is defined as the sum of the number of attributes for all edges and vertices in the graph, specifically: $|\mathcal{G}| = \sum_{e \in E} \#\text{attr}(e) + \sum_{v \in S \cup X} \#\text{attr}(v)$. This normalization accounts for graph size variations, allowing for a more consistent comparison of the impact of changes across different scenarios. By quantifying discrepancies between the updated and original graphs through an analysis of their attribute changes, our approach offers a more precise measurement of changes in the decision-making framework.

By employing these three steps, LP conversion, graph representation, and graph edit distance calculation, we provide a rigorous and systematic approach to quantifying "Decision Information". Since LLMs cannot directly perform this quantification, we utilize them to sense these processes and generate explanatory insights. This approach fills the gap in previous methods that could not quantitatively analyze the impact of changes in constraints. By offering a unified methodology for measuring changes in constraints and parameters, it enables a precise evaluation of information changes. This framework provides a measurable and comparable basis for assessing the impact of these changes on decision-making. It thus anchors the concept of "Decision Information" in both practical and thEORetical contexts, offering more profound insight into how updates shape decisions.

## 4 EXPERIMENTS

### 4.1 EVALUATION BENCHMARK

Despite the availability of numerous open-source datasets in OR, such as NL4OPT (Ramamonjison et al., 2023), ComplexOR (Xiao et al., 2023), NLP4LP (Holzer et al., 2024), and IndustryOR (Tang et al., 2024), these datasets are limited to problem descriptions and are primarily suited for OR modeling needs. There remains a significant gap in datasets specifically tailored for explainable OR, which are crucial for advancing transparency and interpretability in decision-making processes.

To address this issue, we developed a novel benchmark dataset based on the open-source commercial IndustryOR, specifically designed to evaluate explainability in OR tasks. The benchmark includes 30 categorized problems across various domains (e.g., supply chain management, financial investment, logistics management, etc.). Each problem is paired with 10 unique queries that involve diverse or combined modifications to parameters or constraints (e.g., deleting, adding, or updating constraints and parameters). These queries were crafted by OR experts with significant industry experience to ensure both diversity and practical relevance. The dataset's quality and comprehensiveness are validated through iterative expert reviews and comparisons with real-world cases. For every problem, we provide corresponding Python code and employ the Gurobi optimization solver (Bixby, 2007) to determine optimal solutions. Additionally, we also include the ground truth labels for each query for each problem to ensure accurate evaluation. Notably, the question sets and the queries in this benchmark are developed from scratch and managed in-house, guaranteeing that they have not been part of LLM training data, enhancing the robustness of the benchmark for assessing model performance. We provide a complete example in Appendix A.3.

### 4.2 EVALUATION METHODOLOGY

Our evaluation focuses on two aspects: **Modeling Accuracy** and **Explanation Quality**. For the modeling accuracy assessment, we recognize that different code implementations can produce the same optimization results. For instance, two programs solving a linear programming problem may employ different formulations or techniques (e.g., distinct constraint orderings or variable namings), but both can still yield identical optimal solutions. Therefore, rather than directly comparing the generated code to a reference, we evaluate accuracy by comparing the optimization outcomes to ensure correctness and consistency across implementations.

Regarding the explanation quality, although our task is highly specialized and requires expert-level interpretability in OR, we aim to develop an automated evaluation method. Drawing inspiration from (Kondapaneni et al., 2024) and utilizing the capabilities of LLMs for text quality evaluation (Chen et al., 2023; Chiang & Lee, 2023; Hu et al., 2024; Chu et al., 2024; Zytek et al., 2024), we establish expert-crafted templates and use LLMs to assess explanation quality, aiming for a human-level standard. However, this evaluation approach has certain limitations. To address this, we propose two methods in this paper. First, we establish a structured template that specifies key criteria for effective explanations, such as clarity, relevance, and logical coherence. For instance, explanations should explicitly describe the rationale for modifying specific parameters or constraints and explain how these changes affect the optimization results. Second, we conduct a blind review process where OR experts anonymously score the explanations generated by different methods. This approach helps minimize bias, providing a reliable measure of how effectively the explanations convey meaningful insights to users. We will assess whether the proposed automated evaluation method aligns with human evaluation. This dual approach enables us to assess the consistency between the proposed automated method and human evaluation.

### 4.3 BASELINES

We employ two baselines to ensure a comprehensive evaluation: Standard and OptiGuide (Li et al., 2023). The Standard involves a proprietary LLM (e.g., GPT-4, GPT-4-Turbo, etc.) that generates updated programs and explanations, serving as a basic comparison point for assessing overall performance. The OptiGuide represents a specialized method used in supply chain optimization, providing a domain-specific comparison that evaluates the adaptability and effectiveness of LLMs in industry-relevant scenarios.

Table 2: Accuracy across different models under different LLMs with zero/one-shot setting. The bold scores are the best in each row.

| Setting | Model | Standard | OptiGuide | EOR |
|---|---|---|---|---|
| Zero-shot | GPT-4 | 18.67% | 30.33% | **75.67%** |
| | GPT-4-1106-preview | 75.33% | 36.00% | **81.67%** |
| | GPT-4-0125-preview | 68.33% | 47.33% | **87.33%** |
| | GPT-4-Turbo | 63.00% | 30.33% | **88.33%** |
| One-shot | GPT-4 | 71.67% | 75.00% | **90.33%** |
| | GPT-4-1106-preview | 69.67% | 55.33% | **87.67%** |
| | GPT-4-0125-preview | 76.00% | 69.33% | **92.00%** |
| | GPT-4-Turbo | 88.00% | 69.33% | **95.33%** |

## 4.4 MODEL SETUP

For our experiments, we evaluate the performance of the proposed baselines using four LLMs, GPT-4 (Achiam et al., 2023), GPT-4-1106-preview, GPT-4-0125-preview, and GPT-4-Turbo, under both zero-shot and one-shot learning settings. The zero-shot setting requires the models to generate updated programs and explanations without any prior examples, testing their inherent understanding and generalization capabilities. The one-shot setting provides a single example to guide the models, allowing us to assess the impact of minimal contextual information on their ability to perform the tasks accurately and coherently. To ensure fairness and reproducibility, we fix the hyperparameter *temperature* at 0 and apply the same examples for all models in the one-shot setting, and all models are implemented under the framework AutoGen (Wu et al., 2024). This paper focuses on fully automating all processes, excluding user involvement. However, as our implementation is built on the AutoGen, it inherently supports seamless integration of user feedback. A detailed hyperparameter sensitivity analysis is provided in the Appendix A.4. The source code is available at https://github.com/Forrest-Stone/EOR.

## 4.5 QUANTITATIVE PERFORMANCE

### 4.5.1 COMPARISON OF MODELING ACCURACY

Table 2 shows the accuracy results across different models. We have the following observations. In the zero-shot setting, EOR consistently outperforms both Standard and OptiGuide across all LLM versions. These results emphasize the robustness of the EOR in zero-shot tasks, which substantially improves performance compared to other methods. For instance, GPT-4-Turbo achieves 88.33% accuracy with EOR, while Standard and OptiGuide methods yield only 63.00% and 30.33%, respectively. Moreover, for all LLM models except GPT-4, Standard outperforms OptiGuide, indicating that the LLM's modeling capabilities are quite strong.

In the one-shot setting, EOR continues to outperform Standard and OptiGuide in all LLM versions, achieving an average of 90.00% accuracy and even reaching 95.33% accuracy on the GPT-4-Turbo, the highest among all results. Additionally, we find that providing an example can significantly improve modeling accuracy, and nearly all methods perform better in the one-shot setting than in the zero-shot setting. Specifically, the accuracy of OptiGuide on GPT-4 improves from 30.33% to 75.00%. However, OptiGuide still produces the worst result except on GPT-4.

Overall, EOR consistently outperforms other methods in both zero-shot and one-shot settings, particularly with models like GPT-4-Turbo and GPT-4-0125-preview. The accuracy gains observed when transitioning from zero-shot to one-shot highlight the importance of using examples in improving model performance. In terms of model comparisons, GPT-4-Turbo demonstrates the highest adaptability across both settings, achieving the best overall accuracy. While OptiGuide provides modest improvements, it does not match the performance of Standard and EOR. These results underscore the value of carefully selecting both models and example strategies to maximize accuracy, with EOR emerging as the most effective for high-accuracy tasks.

Table 3: Quality scores (0-10) across different models with zero/one-shot setting.

| Method | | EC ↑ | | ER ↑ | | Overall ↑ | |
|---|---|---|---|---|---|---|---|
| | | Auto | Expert | Auto | Expert | Auto | Expert |
| Zero-shot | Standard | 0.12 | - | 6.98 | 6.98 | 5.36 | 5.63 |
| | OptiGuide | 0.69 | - | 7.93 | 7.90 | 6.31 | 6.59 |
| | EOR | 9.76 | 9.86 | 9.41 | 9.47 | 9.47 | 9.47 |
| One-shot | Standard | 0.34 | - | 7.10 | 6.90 | 5.54 | 5.37 |
| | OptiGuide | 0.20 | - | 7.60 | 7.61 | 5.96 | 6.05 |
| | EOR | 9.61 | 9.72 | 9.30 | 9.35 | 9.33 | 9.35 |

### 4.5.2 COMPARISON OF THE QUALITY OF EXPLANATIONS

As discussed in Sec. 4.2, a critical part of our evaluation is assessing the quality of the explanations generated by the models. It is important to note that explanations based on incorrect results are irrelevant, as explaining failure cases offers no meaningful insights. Therefore, we first filter out all incorrect modeling cases, ensuring that only correct outputs are evaluated. We adopt two evaluation methods: an automated evaluation (Auto) and expert evaluation (Expert), both introduced in Sec. 4.2. These evaluations focus on two main aspects: Explanation of Correctness (EC) and Explanation of Results (ER). EC assesses the clarity and correctness of the explanation regarding the model's output, while ER evaluates how well the explanation conveys the reasoning behind the optimization results. Additionally, we also measure the overall explanation quality (Overall). The explanations are scored on a 0-10 scale, with 0 indicating poor quality and 10 indicating excellent quality. The prompt template for evaluating the explanation quality is provided in Appendix A.5. The results of these evaluations using GPT-4o are summarized in Table 3.

As shown in Table 3, we have the following observations. Firstly, Auto is nearly as effective as Expert, with minimal differences in scores across all metrics, indicating that Auto is a reliable and valid approach. It is worth noting that Auto scores individual queries on EC, although it does not explicitly evaluate explanations. Secondly, EOR consistently outperforms both Standard and OptiGuide in explanation quality, achieving the highest scores in both zero-shot and one-shot settings. This demonstrates the superiority of our method in providing clearer and more accurate interpretations compared to other models. Finally, we find that Auto scores slightly higher on Standard than Expert, while for other models, the opposite is observed, with Auto scoring slightly lower than Expert. One possible reason is that LLMs may be biased toward the results they generate.

### 4.6 CASE STUDY

In this section, we provide an example of the code and explanations generated by EOR, as illustrated in Figure 3, where simple explanations are highlighted in green to represent attribute explanations, while deeper ones in yellow indicate justification explanations with quantitative analysis. Results from other models are provided in Appendix A.6. The EOR approach stands out by not only providing a clear explanation of the updated code but also offering detailed insights into how the query affects the outcomes. It explicitly highlights the newly added constraints that limit the number of Type A and Type B aircraft, demonstrating transparency in addressing the operational limits specified in the query. Unlike other methods that merely report result differences, EOR incorporates a quantitative analysis, explaining the $15,000 increase in operational costs due to the restricted solution space caused by the newly added constraints. This step-by-step explanation not only justifies the modifications but also clarifies the broader impact of the constraints, making EOR more comprehensive and contextually aware than other models.

## 5 CONCLUSION AND FUTURE WORK

In this paper, we present EOR, a novel framework that addresses transparency and interpretability challenges in OR. We introduce the concept of "Decision Information" through what-if analysis and use bipartite graphs to quantify changes in OR models. By leveraging LLM capabilities, our

---

**A Case Study of EOR.**

**Query:** How should the aircraft configuration be adjusted to meet demand if the company decides to limit the number of Type A aircraft operated to no more than 15 and the number of Type B aircraft to no more than 30?

- - - - - - - - - - - - - - - - - - - - - - - - - - - - - - - - - - - - - - - - - - - - - - - - -

**The "code" generated by EOR:**

```
{
    "ADD CONSTRAINT": "# Constraint: Limit number of Type A and Type
        B aircraft\nm.addConstr(aircraft_count['A'] <= 15, name
        =\"MaxTypeAConstraint\")\nm.addConstr(aircraft_count['B']
        <= 30, name=\"MaxTypeBConstraint\")"
}
```

- - - - - - - - - - - - - - - - - - - - - - - - - - - - - - - - - - - - - - - - - - - - - - - - -

**The explanations generated by EOR:** **Explanation of Updated Code:**
The updated code includes new constraints that implement operational limits on the number of Type A and Type B aircraft the airline can use. Specifically:
- A constraint has been added to limit the number of Type A aircraft to 15.
- A constraint has been added to limit the number of Type B aircraft to 30.
These additional constraints are motivated by possibly new operational limitations or strategic decisions, such as maintenance availability or cost considerations. By imposing these restrictions, the model must now find an optimal solution within a more constrained operational environment.

**Explanation of the Query on Results:**
The query specifically asked to adjust aircraft configurations within newly defined limits. The original model found an optimal solution at a minimum cost of $200,000. However, with the added constraints, the solution space is reduced - fewer configurations of aircraft are available to meet the passenger demand.
As a result of these constraints: The new optimal cost calculated by the model rose to $215,000. This increment indicates that the more stringent operational limits forced the use of a costlier configuration. The query's impact on the results is substantial, as illustrated by a $15,000 increase in operational costs. If rated on a scale from 1 to 10, the impact could be quantified around an 8. This rating reflects significant changes in operational costs and constraints.
The numerical changes in the model, calculated as 100, follows from a step-by-step process analyzing the modifications in the linear programming representation and their effects on the resulting computational graph. This method, grounded in operations research analysis, validates the substantial impact of the query-driven modifications.
In summary, the changes to the code (limiting aircraft types) constrained the model significantly, leading to a more expensive operational solution. The rationale behind integrating these specific constraints could be associated with real-world operational limits, reflecting the complex decision-making processes airlines undergo to balance cost, capacity, and operational feasibility.

Figure 3: An example illustrating the code and explanations generated by EOR.

framework enhances the modeling process and generates actionable, comprehensible explanations. Furthermore, we establish the first industrial benchmark for evaluating explanation quality in OR. In the future, we aim to develop more accurate modeling methods to improve the accuracy while exploring effective automated evaluation techniques for explanations.

ACKNOWLEDGMENTS

This work was supported by the Early Career Scheme (No. CityU 21219323) and the General Research Fund (No. CityU 11220324) of the University Grants Committee (UGC), and the NSFC Young Scientists Fund (No. 9240127).

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

# A APPENDIX

## A.1 THE COMPARISON BETWEEN WHAT-IF ANALYSIS, SENSITIVITY ANALYSIS, AND DECISION INFORMATION ANALYSIS

**What-if Analysis:** What-if analysis explores the impact of changing conditions, including parameters, on overall outcomes, allowing users to adjust inputs to observe potential scenarios manually. This approach emphasizes scenario exploration at a macro level, helping users understand how different "what if" situations might influence results without delving into specific details.

**Sensitivity Analysis:** Sensitivity analysis measures how minor variations in input parameters affect the model's output, identifying which inputs have the most significant influence on results. This method provides detailed quantitative insights into the effects of parameter changes, making it particularly effective in optimization contexts where analytical solutions can be derived.

**Decision Information Analysis:** Decision information analysis extends sensitivity analysis by focusing on how changes in constraints impact decision-making and outcomes. This approach examines the sensitivity of the model to variations in constraints, identifying key decision factors while also capturing the broader implications of both constraint and parameter changes, particularly in complex optimization problems without analytical solutions.

In summary, decision information and sensitivity analysis are more detailed, with the former centering on decision-related changes (constraints or parameters) and the latter on parameter sensitivity, while both can be part of a broader what-if analysis framework. Previous research has predominantly focused on sensitivity analysis through the simplex method, which is recognized for its efficiency, precision, and ability to provide analytical solutions. However, the simplex method has limitations in evaluating the effects of changes to constraints. In contrast, our proposed approach leverages bipartite graphs to assess the impact of constraint modifications, addressing this gap and providing a more generalized solution.

## A.2 PROMPT TEMPLATE DESIGN

### A.2.1 PROMPT TEMPLATE FOR WRITER AGENT

The prompt template for Writer with system message for the ChatCompletion inference:

```
WRITER_SYSTEM_MSG = """

**Role:** You are a chatbot tasked with:
(1) Writing Python code for operations research-related projects.
(2) Explaining solutions using the Gurobi Python solver.

--- Problem Description: ---
{description}

--- Source Code: ---
{source_code}

--- Documentation: ---
{doc_str}

--- Example Q&A: ---
{example_qa}

--- Original Execution Result: ---
{execution_result}

**Task:**
You are provided with the original problem description and the correct
    ↪ code for an operations research problem. Based on the user's query,
    ↪  update the code accordingly. Your task may involve either deleting
    ↪  constraints or adding new data or constraints.
```

```
25
26
27  **Steps:**
28
29  1. Determine the Required Operations:
30
31     - **Add Operation:** Generate new code for data or constraints to be
          ↪ added.
32       - Insert the new data between the markers:
33         "# EORer DATA CODE GOES HERE" and "# EORer DATA CODE ENDS HERE".
34       - Insert the new constraints between the markers:
35         "# EORer CONSTRAINT CODE GOES HERE" and "# EORer CONSTRAINT CODE
              ↪ ENDS HERE".
36
37     - **Delete Operation:** Identify the relevant block within the
          ↪ constraints section that needs to be deleted.
38       - The code to be deleted will be between the markers:
39         "# EORer CONSTRAINT CODE GOES HERE" and "# EORer CONSTRAINT CODE
              ↪ ENDS HERE".
40
41  2. Return the Changes in JSON Format:
42
43     - Use the keys "DELETE CONSTRAINT", "ADD CONSTRAINT", or "ADD DATA".
          ↪ Only these keys are allowed.
44     - The values should be the Python code snippet to be deleted (exactly
          ↪ as it appears in the original code) or the new code to be added.
45     - Include comments within the code snippets using the prefix "#".
46     - Ensure the line breaks and indents of the code are correct.
47
48  3. Output Requirements:
49
50     - Return only the JSON object with the changes.
51     - Do not include any additional information in the response.
52     - Do not add new decision variables.
53     - Ensure the JSON is valid, properly formatted.
54
55
56  The above explained instructions are your guide to accomplish the task
        ↪ effectively. Your user's success heavily relies upon your ability
        ↪ to provide the precise and accurate Python code changes within the
        ↪ existing operations research problem. Good Luck!
57
58  """
```

The prompt template for Code (2):

```
1   CODE_PROMPT = """
2
3   **Role:** You are a professional software developer tasked with handling
        ↪ code modification requests. Your role is to interpret these
        ↪ requests which describe changes needed in a source code.
4
5
6   **Task:** Your task is to return the changes to be made to the source
        ↪ code based on the user's query. The modifications should be
        ↪ returned in a JSON format containing only the necessary changes.
7
8
9   **JSON Format Requirements:**
10
11     - Use the keys "DELETE CONSTRAINT", "ADD CONSTRAINT", or "ADD DATA".
          ↪ Only these keys are allowed.
12     - The values should be the Python code snippet to be deleted (exactly
          ↪ as it appears in the original code) or the new code to be added.
13     - Include comments within the code snippets using the prefix "#".
```

```
14      - Ensure the line breaks and indents of the code are correct.
15
16
17   **Output Requirements:**
18
19      - Return only the JSON object with the changes.
20      - Do not include any additional information in the response.
21      - Do not add new decision variables.
22      - Ensure the JSON is valid, properly formatted.
23
24
25   The above explained instructions are your guide to accomplish the task
         ↪ effectively. Your user's success heavily relies upon your ability
         ↪ to provide the precise and accurate Python code changes within the
         ↪ existing operations research problem. Good Luck!
26
27
28   --- Answer Code: ---
29
30   """
```

The prompt template for Debug (2):

```
1    DEBUG_PROMPT = """
2
3    **Role:** You are a professional code debugger.
4
5
6    **Task:** Identify and fix the error in the code, ensuring the corrected
         ↪ version runs smoothly and error-free.
7
8
9    **Details:**
10      - Error Type: {error_type}
11      - Error Message: {error_message}
12
13
14   **Instructions:**
15   Please analyze the error details, resolve the bug based on the type and
         ↪ message provided, and rewrite the corrected version of the code
         ↪ snippet below.
16
17
18   **Corrected Code:**
19   --- NEW CODE ---
20
21   """
```

The prompt template for Interpreter (6):

```
1    INTERPRETER_PROMPT = """
2
3    **Role:** You are a skilled interpreter with expertise in analyzing and
         ↪ explaining changes in computational model code and their effects on
         ↪  results.
4
5
6    **Task:** Present a clear and thorough explanation of the code updates
         ↪ and their effects on the results. Structure your explanation in two
         ↪  key parts: Explanation of the updated code and Explanation of the
         ↪ Query on Results.
7
8
9    **Inputs:**
```

```
10      - Original code: {source_code}
11      - Updated code: {new_code}
12      - Code changes induced by the query: {json_data}
13      - Original execution results: {original_execution_result}
14      - New execution results: {execution_rst}
15      - Measure of numerical changes in the model induced by the query: {
           ↪ different_model}
16
17
18  **Key Points to Understand:**
19
20  1. Explanation of Updated code:
21
22      - Explain the rationale behind each specific change to the code, such
           ↪ as why certain constraints or data were added, deleted, or
           ↪ modified.
23
24  2. Explanation of the Query on Results:
25
26      - Clarify why the specific results were produced in response to the
           ↪ query.
27      - Assess the query's impact on the results by comparing the new
           ↪ execution results with the original ones and the corresponding
           ↪ numerical changes in the model.
28      - Use a scale from 1 to 10 to quantify the query's impact on the
           ↪ results, with 1 indicating minimal impact and 10 indicating
           ↪ significant impact.
29
30
31  **Background for Numerical Changes Calculation:**
32
33  The impact of the query is measured using a three-step process:
34      1. LP Conversion: The problem is converted into a linear programming (
           ↪ LP) format to identify key components and constraints.
35      2. Graph Representation: The LP model is then represented as a
           ↪ bipartite graph, where nodes and edges correspond to variables,
           ↪ constraints, and relationships.
36      3. Graph Edit Distance Calculation: The difference between the
           ↪ original and modified graphs is computed by measuring the graph
           ↪ edit distance, which involves operations like insertion,
           ↪ deletion, and substitution of nodes and edges, each with a unit
           ↪ cost of 1.
37
38
39  **Output:**
40
41  Provide the explanations in two distinct parts:
42      (1) Explanation of the Updated code
43      (2) Explanation of the Query on Results
44
45
46  **Requirements:**
47
48      - Ensure the explanations are detailed and comprehensive, covering all
           ↪  relevant aspects of the code updates and their impact on the
           ↪ results.
49      - Ensure that explanations are delivered in a narrative style, suited
           ↪ for a non-technical audience, avoiding jargon or direct
           ↪ references to specific variable names.
50      - Offer clear, precise, easy-to-understand descriptions that
           ↪ effectively bridge complex information with clarity and insight.
51
52
53  The above explained instructions are your guide to accomplish the task
       ↪ effectively. Your user's success heavily relies upon your ability
```

```
      ↪ to provide the explanations within the existing operations research
      ↪  problem. Good Luck!
54
55
56 --- HUMAN READABLE ANSWER ---
57
58 """
```

### A.2.2 PROMPT TEMPLATE FOR SAFEGUARD AGENT

The prompt template for Safeguard with system message for the ChatCompletion inference:

```
1  SAFEGUARD_SYSTEM_MSG = """
2
3  **Role:** You are a code safety evaluator.
4
5
6  **Task:** Review the provided source code to determine if it is safe to
      ↪ execute, ensuring it does not contain any malicious code that could
      ↪  compromise security or privacy.
7
8
9  **Instructions:**
10
11 --- Source Code: ---
12 {source_code}
13
14 **Question:**
15 Is the code safe to run?
16
17 **Answer:**
18 Respond with one word:
19    `SAFE` if the code is secure.
20    `DANGER` if the code poses any risk.
21
22 """
```

The prompt template for Safe (4):

```
1  SAFEGUARD_PROMPT = """
2
3  **Role:** You are a professional code safety evaluator.
4
5
6  **Task:** Examine the safety of each code snippet contained within a
      ↪ provided JSON file.
7
8
9  **Details:**
10
11    - **Code Structured as JSON:** Each value in the JSON represents a
          ↪ code snippet intended for review. These snippets may be newly
          ↪ generated or under consideration for deletion.
12
13
14 **Instructions:**
15
16    - Thoroughly analyze each code snippet found in the JSON.
17    - For each snippet, determine its safety for execution.
18    - Provide your assessment as a single word for each snippet: either "
          ↪ SAFE" or "DANGER".
19
20
```

```
21   **Example of Expected Response:** For snippet_1: SAFE, for snippet_2:
     ↪ DANGER
22
23
24   --- Answer: ---
25
26   """
```

## A.3   AN EXAMPLE OF THE BENCHMARK

The original problem description and the result of this problem:

```
1   Problem Description:
2   An airline operates two types of aircraft: large aircraft (Type A) and small aircraft (Type B). Each
        ↪ type of aircraft has different operating costs and passenger capacities. The company needs
        ↪  to determine the number of each type of aircraft to operate in order to meet the demand
        ↪ of transporting at least 10,000 passengers. Specifically, one Type A aircraft can carry 500
        ↪ passengers, and one Type B aircraft can carry 200 passengers. However, due to the use
        ↪ and maintenance requirements of the aircraft, the total number of Type A and Type B
        ↪ aircraft operated cannot exceed 50. The cost of operating one Type A aircraft is $10,000,
        ↪ and the cost trgtgof operating one Type B aircraft is $5,000. Due to the indivisibility of
        ↪ the aircraft, both types of aircraft must be operated in integer quantities. Under these
        ↪ conditions, what is the minimum total cost that satisfies the passenger transportation
        ↪ demand while adhering to the operational restrictions? Please round the answer to the
        ↪ nearest integer.
3
4   Problem Result:
5   200000.0
```

The original Python code with Gurobi:

```python
1   import gurobipy as gp
2   from gurobipy import GRB
3
4
5   # Parameters Section Begin
6   # Define model parameters
7   aircraft_types = ['A', 'B']
8
9   # Passenger capacity per aircraft type
10  passenger_capacity = {
11      'A': 500,
12      'B': 200
13  }
14
15  # Operating cost per aircraft type (dollars)
16  operating_cost_per_aircraft = {
17      'A': 10000,
18      'B': 5000
19  }
20
21  # Minimum passenger demand
22  min_passenger_demand = 10000
23
24  # Maximum number of aircraft
25  max_aircraft_count = 50
26  # Parameters Section End
27
28
29  # EORer DATA CODE GOES HERE
30
31
```

```python
32  # EORer DATA CODE ENDS HERE
33
34
35  # Create a Gurobi model
36  m = gp.Model("AirlineOptimization")
37
38  # Decision Variables Section Begin
39  # Create decision variables a and b for the number of large and small
        ↪ aircraft respectively
40  aircraft_count = {
41      'A': m.addVar(vtype=GRB.INTEGER, name="aircraft_A"),
42      'B': m.addVar(vtype=GRB.INTEGER, name="aircraft_B")
43  }
44  # Decision Variables Section End
45
46
47  # Objective Function Section Begin
48  # Set the objective function to minimize total operating cost
49  m.setObjective(
50      gp.quicksum(
51          operating_cost_per_aircraft[t] * aircraft_count[t]
52          for t in aircraft_types
53      ),
54      sense=GRB.MINIMIZE
55  )
56  # Objective Function Section End
57
58
59  # EORer CONSTRAINTS CODE GOES HERE
60
61
62  # Constraints Section Begin
63  # Constraint: Meet the passenger transportation demand
64  m.addConstr(
65      gp.quicksum(passenger_capacity[t] * aircraft_count[t]
66              for t in aircraft_types) >= min_passenger_demand,
67      name="PassengerDemandConstraint"
68  )
69
70  # Constraint: The total number of aircraft cannot exceed the maximum
        ↪ allowed
71  m.addConstr(
72      gp.quicksum(aircraft_count[t]
73              for t in aircraft_types) <= max_aircraft_count,
74      name="OperationalConstraint"
75  )
76  # Constraints Section End
77
78
79  # EORer CONSTRAINTS CODE MIDDLE HERE
80
81
82  # EORer CONSTRAINTS CODE ENDS HERE
83
84
85  # Solving the Model Section Begin
86  # Solve the model
87  m.optimize()
88
89  # Output the results
90  if m.status == GRB.OPTIMAL:
91      print(f"Minimum total cost: {round(m.ObjVal)} dollars")
92      for t in aircraft_types:
93          print(f"Number of Type {t} aircraft: {aircraft_count[t].X}")
94  else:
```

```
95      print("No optimal solution found.")
96  # Solving the Model Section End
```

The queries and truth labels of these updated problems with new queries:

```
1   Query 1: How should the number of aircraft be adjusted to maximize economic efficiency if the
          ↪ operating cost of the large aircraft (Type A) is reduced to $8,000?
2   Truth Label 1: 160000.0
3
4   Query 2: How should the number of aircraft be reassessed to meet transportation demand if the
          ↪ passenger capacity of the small aircraft (Type B) increases to 250 passengers?
5   Truth Label 2: 200000.0
6
7   Query 3: How should the aircraft configuration be adjusted to maximize profit if the operating cost
          ↪ of the Type B aircraft decreases to $4,000 and the passenger capacity of the Type A
          ↪ aircraft increases to 550 passengers?
8   Truth Label 3: 184000.0
9
10  Query 4: How should the number of aircraft be adjusted to maintain minimum total cost if the
          ↪ operating costs of both Type A and Type B aircraft increase by 10%?
11  Truth Label 4: 220000.0
12
13  Query 5: How should the aircraft configuration be adjusted to meet demand if the company decides
          ↪  to limit the number of Type A aircraft operated to no more than 15 and the number of
          ↪ Type B aircraft to no more than 30?
14  Truth Label 5: 215000.0
15
16  Query 6: How should the number of aircraft be adjusted to maintain demand if the passenger
          ↪ capacity of the Type A aircraft decreases to 450 passengers and the operating cost of the
          ↪ Type B aircraft increases to $6,000?
17  Truth Label 6: 226000.0
18
19  Query 7: How should the number of aircraft be adjusted to maximize transportation efficiency if
          ↪ the passenger capacity of the large aircraft increases to 600 passengers and the cost of the
          ↪ small aircraft increases to $5,500?
20  Truth Label 7: 170000.0
21
22  Query 8: How should the number of aircraft be arranged to meet passenger demand and minimize
          ↪ costs if the airline needs to operate at least 10 Type B aircraft?
23  Truth Label 8: 210000.0
24
25  Query 9: How should the number of aircraft be adjusted to maintain economic efficiency if the
          ↪ passenger capacity of the small aircraft (Type B) decreases to 150 passengers and the cost
          ↪ of the large aircraft (Type A) increases to $12,000?
26  Truth Label 9: 240000.0
27
28  Query 10: How will the optimal aircraft configuration change if the constraint that the total number
          ↪  of Type A and Type B aircraft operated cannot exceed 50 is removed?
29  Truth Label 10: 200000.0
```

## A.4  HYPERPARAMETER SENSITIVITY ANALYSIS

This section analyzes the sensitivity of two hyperparameters, *temperature* and *debug times*, to eval-uate their impact on the model's reliability and stability.The *temperature* reflects the reliability of the model's outputs, while *debug times* assess its performance stability. The experimental results are summarized in Tables 4 and 5.

Table 4 shows that our model maintains reliable outputs across temperature settings of 0, 0.5, and 1 in both zero-shot and one-shot scenarios. This demonstrates the model's robustness in generat-

ing consistent and precise outputs under varying temperature configurations, which is particularly important for tasks demanding factual accuracy, such as OR modeling.

Table 5 reveals that increasing the number of debug attempts from 3 to 10 does not significantly improve performance. Additional iterations primarily consume resources without yielding better results, reflecting the model's limited self-correction capabilities. This suggests that external interventions, such as user feedback, may be necessary to enhance performance further.

In summary, the model exhibits strong reliability across different temperature settings and consistent stability regardless of the number of debugging attempts. These findings underscore the robustness of our approach in delivering reliable and stable performance.

Table 4: Accuracy under different methods in zero/one-shot setting with different *temperature*. The bold scores are the best in each row.

| Method | | Temperature | | |
|---|---|---|---|---|
| | | 0 | 0.5 | 1 |
| Zero-shot | OptiGuide | **30.33%** | 22.00% | 26.33% |
| | EOR | 88.33% | **89.67%** | 88.00% |
| One-shot | OptiGuide | 69.33% | **70.00%** | 68.33% |
| | EOR | **95.33%** | 93.67% | 93.33% |

Table 5: Accuracy under different LLMs in zero/one-shot setting with different *debug times*. The bold scores are the best in each row.

| Method | | Debug times | |
|---|---|---|---|
| | | 3 | 10 |
| Zero-shot | GPT-4-1106-preview | **81.67%** | 81.33% |
| | GPT-4-Turbo | 88.33% | **89.00%** |
| One-shot | GPT-4-1106-preview | **87.67%** | **87.67%** |
| | GPT-4-Turbo | **95.33%** | **95.33%** |

## A.5   PROMPT TEMPLATE FOR EVALUATING THE EXPLANATIONS

```
EXPLANATIONS_EVALUATION = """

**Role:** You are an expert in Operations Research evaluating
    ↪ explanations provided by three different models: `EOR`, `OptiGuide
    ↪ `, and `Standard`. Your role is to assess the quality of these
    ↪ explanations based on a user query.

**Input:**
    - User Query: {query}
    - Explanations by EOR: {EOR}
    - Explanations by OptiGuide: {optiguide}
    - Explanations by Standard: {standard}

**Task:** For each model, provide three scores:

    1). Score 1 - Explanation of Updated Code:
    How clearly does the explanation describe the code modifications made
        ↪ in response to the query? If no explanation of code updates is
        ↪ provided, score 0.

    2). Score 2 - Explanation of Query on Results:
    How well does the explanation clarify why and how the results changed
        ↪ due to the query? Focus on the depth of the explanation,
```

```
              ↪ particularly the quantitative reasoning behind the changes, not
              ↪ just a description of the result.
20
21    3). Score 3 - Overall Score:
22    Based on the previous scores, assign an overall score (0-10)
              ↪ reflecting the combined quality and effectiveness of the
              ↪ explanation.
23
24
25 **Scoring Criteria:**
26
27    - Scores should range from 0 (poor) to 10 (excellent) for each
              ↪ category.
28    - Consider the overall clarity, conciseness, and structure. The
              ↪ explanation should be easy to follow and understand.
29
30
31 **Output JSON Format:**
32
33    - Only return the results in the JSON format.
34    - The keys should be `EOR`, `OptiGuide`, and `Standard`.
35    - Each key should have a list with the three scores (0-10).
36    - Do not include any additional information or comments in the
              ↪ response.
37
38
39 Your evaluation will help determine which model provides the most
       ↪ effective and clear explanations for the given query. Good Luck!
40
41
42 --- Answer: ---
43
44 """
```

### A.6  CASE STUDY

The comparison of code and explanations generated by different models are shown in Figure 4 and 5, respectively.

As shown in Figure 4, EOR returns a code snippet in JSON format, OptiGuide provides a code snippet, and Standard returns the complete code for the entire problem. Although Standard achieves better experimental results than OptiGuide, its modification and return of the entire code significantly increases maintenance costs, making it challenging to track changes. OptiGuide provides markers to guide code modification in larger models, but experiments have shown that these markers are sometimes removed, complicating efforts to identify what and where changes were made. In contrast, the code format proposed in our paper enables targeted updates at specific locations while ensuring markers remain intact. This not only reduces maintenance costs but also, when paired with clear explanations, simplifies the identification of both the location and nature of code updates.

As shown in Figure 5, both Standard and OptiGuide generate explanations that mainly describe results and provide a brief analysis of changes in solution outcomes. Still, they lack quantitative analysis, especially concerning the impact of implicit query changes on the model. These explanations align more closely with the attribution explanation, as defined in our paper. In contrast, our approach emphasizes the justification explanation, which not only explains and analyzes code updates but also provides a quantitative analysis of the impact of constraint changes on the model.

### A.7  FAILURE CASES

To understand the strengths and weaknesses of EOR, we analyzed common failure cases, summarized in Table 6. These failures can be grouped into three categories: JSON format errors, where the LLMs fail to generate correct JSON outputs; correct execution, where code runs but produces incorrect results due to modeling logic errors or incomplete modeling; and runtime errors, such as

variable name errors, syntax errors, or indentation errors. Table 6 shows a 60.00% reduction in total errors from zero-shot to one-shot, demonstrating a substantial improvement in the model's performance. No syntax errors in the one-shot results highlights the effectiveness of EOR. However, ongoing attention to modeling logic and runtime errors is still crucial for further improvement.

Table 6: Failure cases on GPT-4-Turbo with zero/one-shot setting.

| Failure Types | | Zero-shot (Total 35) | | One-shot (Total 14) | |
|---|---|---|---|---|---|
| | | #Number | Percentage | #Number | Percentage |
| JSON Format Errors | | 8 | 22.86% | 2 | 14.29% |
| Correct Execution | Modeling Logic Errors | **13** | **37.14%** | **4** | **28.57%** |
| | Incomplete Modeling | 4 | 11.43% | 2 | 14.29% |
| Runtime Errors | Variable Name Errors | 2 | 5.71% | 3 | 21.43% |
| | Syntax Errors | 6 | 17.14% | 0 | 0.00% |
| | Indent Errors | 2 | 5.71% | 3 | 21.43% |

---

**Comparison of Code Generated by Different Models.**

**Query:** How should the aircraft configuration be adjusted to meet demand if the company decides to limit the number of Type A aircraft operated to no more than 15 and the number of Type B aircraft to no more than 30?

- - - - - - - - - - - - - - - - - - - - - - - - - - - - - - - - - - - - - - - - - - - - - - - - - - -

**EOR:**

```
1  {
2      "ADD CONSTRAINT": "# Constraint: Limit number of Type A and Type
         ↪  B aircraft\nm.addConstr(aircraft_count['A'] <= 15, name
         ↪ =\"MaxTypeAConstraint\")\nm.addConstr(aircraft_count['B']
         ↪ <= 30, name=\"MaxTypeBConstraint\")"
3  }
```

- - - - - - - - - - - - - - - - - - - - - - - - - - - - - - - - - - - - - - - - - - - - - - - - - - -

**OptiGuide:**

```
1  "# Additional constraints for the maximum number of Type A and Type
       ↪  B aircraft\nm.addConstr(aircraft_count['A'] <= 15, name=\"
       ↪ MaxAircraft_A\")\nm.addConstr(aircraft_count['B'] <= 30, name
       ↪ =\"MaxAircraft_B\")"
```

**Standard:**

```
1  "import gurobipy as gp\nfrom gurobipy import GRB\n\n# Parameters
       ↪ Section Begin\n# Define model parameters\naircraft_types = ['
       ↪ A', 'B']\n\n# Passenger capacity per aircraft type\
       ↪ npassenger_capacity = {\n 'A': 500,\n 'B': 200\n}\n\n#
       ↪ Operating cost per aircraft type (dollars)\
       ↪ noperating_cost_per_aircraft = {\n 'A': 10000,\n 'B': 5000\n
       ↪ }\n\n# Minimum passenger demand\nmin_passenger_demand =
       ↪ 10000\n\n# Maximum number of aircraft for Type A and Type B\
       ↪ nmax_aircraft_count_A = 15\nmax_aircraft_count_B = 30\n#
       ↪ Parameters Section End\n\n# Create a new Gurobi model\nm = gp
       ↪ .Model(\"AirlineRevisedOptimization\")\n\n# Decision
       ↪ Variables Section Begin\n# Create decision variables for the
       ↪ number of large and small aircraft respectively\
       ↪ naircraft_count = {\n 'A': m.addVar(vtype=GRB.INTEGER, name
       ↪ =\"aircraft_A\"),\n 'B': m.addVar(vtype=GRB.INTEGER, name=\"
       ↪ aircraft_B\")\n}\n# Decision Variables Section End\n\n#
       ↪ Objective Function Section Begin\n# Set the objective
       ↪ function to minimize total operating cost\nm.setObjective(\n
       ↪ gp.quicksum(\n operating_cost_per_aircraft[t] *
       ↪ aircraft_count[t]\n for t in aircraft_types\n ),\n sense=GRB.
       ↪ MINIMIZE\n)\n# Objective Function Section End\n\n#
       ↪ Constraints Section Begin\n# Constraint: Meet the passenger
       ↪ transportation demand\nm.addConstr(\n gp.quicksum(
       ↪ passenger_capacity[t] * aircraft_count[t]\n for t in
       ↪ aircraft_types) >= min_passenger_demand,\n name=\"
       ↪ PassengerDemandConstraint\"\n)\n\n# Constraint: The number of
       ↪  Type A aircraft cannot exceed 15\nm.addConstr(\n
       ↪ aircraft_count['A'] <= max_aircraft_count_A,\n name=\"
       ↪ TypeAConstraint\"\n)\n\n# Constraint: The number of Type B
       ↪ aircraft cannot exceed 30\nm.addConstr(\n aircraft_count['B']
       ↪  <= max_aircraft_count_B,\n name=\"TypeBConstraint\"\n)\n#
       ↪ Constraints Section End\n\n# Solving the Model Section Begin\
       ↪ n# Solve the model\nm.optimize()\n\n# Output the results\nif
       ↪ m.status == GRB.OPTIMAL:\n print(f\"Optimized minimal total
       ↪ cost with new configuration: {round(m.ObjVal)} dollars\")\n
       ↪ for t in aircraft_types:\n print(f\"Number of Type {t}
       ↪ aircraft: {aircraft_count[t].X}\")\nelse:\n print(\"No
       ↪ optimal solution found.\")\n# Solving the Model Section End"
```

Figure 4: An example illustrating codes generated by different models.

---

**Comparison of Explanations by Different Models.**

**Query:** How should the aircraft configuration be adjusted to meet demand if the company decides to limit the number of Type A aircraft operated to no more than 15 and the number of Type B aircraft to no more than 30?

- - - - - - - - - - - - - - - - - - - - - - - - - - - - - - - - - - - - - - - - - - - - - - - - - - - -

**EOR:** \*\*Explanation of Updated Code:\*\*
The updated code includes new constraints that implement operational limits on the number of Type A and Type B aircraft the airline can use. Specifically:
    - A constraint has been added to limit the number of Type A aircraft to 15.
    - A constraint has been added to limit the number of Type B aircraft to 30.
These additional constraints are motivated by possibly new operational limitations or strategic decisions, such as maintenance availability or cost considerations. By imposing these restrictions, the model must now find an optimal solution within a more constrained operational environment.

\*\*Explanation of the Query on Results:\*\*
The query specifically asked to adjust aircraft configurations within newly defined limits. The original model found an optimal solution at a minimum cost of $200,000. However, with the added constraints, the solution space is reduced - fewer configurations of aircraft are available to meet the passenger demand.
As a result of these constraints: The new optimal cost calculated by the model rose to $215,000. This increment indicates that the more stringent operational limits forced the use of a costlier configuration. The query's impact on the results is substantial, as illustrated by a $15,000 increase in operational costs. If rated on a scale from 1 to 10, the impact could be quantified around an 8. This rating reflects significant changes in operational costs and constraints.
The numerical changes in the model, calculated as 100, follows from a step-by-step process analyzing the modifications in the linear programming representation and their effects on the resulting computational graph. This method, grounded in operations research analysis, validates the substantial impact of the query-driven modifications.
In summary, the changes to the code (limiting aircraft types) constrained the model significantly, leading to a more expensive operational solution. The rationale behind integrating these specific constraints could be associated with real-world operational limits, reflecting the complex decision-making processes airlines undergo to balance cost, capacity, and operational feasibility.

- - - - - - - - - - - - - - - - - - - - - - - - - - - - - - - - - - - - - - - - - - - - - - - - - - - -

**OptiGuide:** The optimization problem was re-solved with additional constraints limiting the number of Type A aircraft to no more than 15 and Type B aircraft to no more than 30. The new configuration resulted in a total operating cost of $215,000. When compared to the original solution, where the total operating cost was $200,000, this new configuration increased the operating costs by $15,000 due to the more restrictive constraints on the number and types of aircraft deployed. This demonstrates the trade-offs between operational flexibility and cost.

- - - - - - - - - - - - - - - - - - - - - - - - - - - - - - - - - - - - - - - - - - - - - - - - - - - -

**Standard:** Based on the updated constraints where the maximum number of Type A aircraft is limited to 15 and Type B aircraft to 30, the revised optimization solution results in a total operating cost of $215,000. This is an increase from the original scenario, due to the restrictions on the number of aircraft which limited the company's ability to opt for the least expensive configuration to meet passenger demand. In this setup, the solution ensures passenger requirements are met while respecting the new operational limits placed on the aircraft types.

---

Figure 5: An example illustrating explanations generated by different models.

