# OpenReview forum: "Decision Information Meets Large Language Models: The Future of Explainable Operations Research"
_ICLR.cc/2025/Conference — ICLR 2025 Poster_

### Official Review · Reviewer_iUYd · 2024-10-25

**Soundness:** 3
**Presentation:** 3
**Contribution:** 3
**Rating:** 8
**Confidence:** 2

**Summary:**

This paper proposes a framework called Explainable Operations Research (EOR) that combines Operation Research (OR) with LLMs. This idea aims to improve the clarity and trustworthiness of decision-making processes. The ultimately expected results should be clear, actionable explanations of how changes in constraints or parameters would affect OR solutions.

The claimed contributions in this paper are: (1) the authors formulate the problem of the explainable OR problems within the context of LLMs, (2) the authors introduce the concept of "Decision Information" and utilize bipartite graphs with LLMs to quantify the importance, (3) the authors develop a new benchmark specifically designed to evaluate the effectiveness.

**Strengths:**

Although I am not deeply familiar with the OR domain, the integration of LLMs with OR problems seems both interesting and innovative. While I initially had concerns about the strict reasoning required for OR problems, it appears that these concerns are somehow addressed through the design choices and the experimental results presented in the paper. However, I may have missed some of the finer details of the concepts.

Overall, the paper is well-presented and easy to follow. Some figures and diagrams could be simplified or made more intuitive, which would help readers with less background in LLMs or NLP better understand the concepts. In addition to the methods, the experimental setup is convincing and thorough.

The significance of this approach is clear. By using this framework, users can gain insights into critical aspects of OR problems that have often been overlooked, enhancing their understanding and decision-making.

**Weaknesses:**

Since developing a benchmark is listed as one of the contributions, a more detailed introduction to this benchmark is expected. It would be helpful to expand on things like why certain problem categories were selected, how comprehensive the benchmark is across various OR tasks, and whether it reflects real-world complexity. Furthermore, examples showing how the benchmark effectively measures explainability in practice would strengthen the paper. This would provide more transparency about how the benchmark can be applied and generalized.

Although the paper presents a dual evaluation (automated and expert) to assess explanation quality, it doesn’t fully explore how well these metrics align with user needs (who are, in other words, the major user groups if it is put into real-world use). For instance, it is unclear whether the explanation quality metrics are used to capture all the necessary aspects of understandability, particularly for non-experts. Providing more insights into how the explanations are judged, probably by including user feedback or more detailed breakdowns of the evaluation criteria, would offer a stronger case for the robustness of the framework.

An analysis of computation complexity/scalability would also help strengthen the paper. I have potential concerns regarding the computational complexity of the proposed framework, especially given the use of LLMs and graph-based methods.

**Questions:**

See some of the questions above in Weakness. In addition:

Regarding the benchmark (for explainability): What were the criteria for selecting the problem categories, and how comprehensive is the benchmark in terms of covering different types of OR challenges? Additionally, have you validated the benchmark with industry professionals or in practical settings to ensure it reflects real-world complexity?

Regarding the system-level workflow: Have you considered how decision-makers would interact with the system and its explanations in real time? What if the system makes mistakes, can it be corrected under human supervision?

Regarding the experimental setups/scalability: The paper compares EOR to OptiGuide, but have you considered comparing your work to other explainability frameworks outside the OR domain? Can this work be somehow extended to other similar domains but with different contexts, e.g., automated reasoning?

---

> ### Author Response · Authors · 2024-11-20
> **Thanks for your comment!**
>
> We sincerely appreciate the constructive suggestions and comments on our work, which have definitely helped us to enhance the paper. The following are our detailed responses to each of the reviewer's comments.
>
> ### **W1: Benchmark Development:**
>
> **A more detailed introduction to this benchmark is expected. It would be helpful to expand on things like why certain problem categories were selected, how comprehensive the benchmark is across various OR tasks, and whether it reflects real-world complexity.**
>
> This benchmark is built on the real-world open-source commercial dataset, IndustryOR [1], ensuring practical relevance. The selected problem categories, such as supply chain management, financial investment, and logistics management, were chosen for their widespread industry impact and ability to represent diverse and critical OR challenges. This ensures the benchmark comprehensively covers key OR tasks.
>
> To reflect real-world complexity, three OR experts with PhDs and significant industry experience developed 300 realistic queries. These queries simulate operational challenges by introducing variations such as resource constraints, and dynamic rules. Iterative validation through expert reviews and real-case comparisons ensures quality and relevance.
>
> We will expand the paper to detail the benchmark rationale, comprehensiveness, and how it captures real-world complexities.
>
> **Furthermore, examples showing how the benchmark effectively measures explainability in practice would strengthen the paper. This would provide more transparency about how the benchmark can be applied and generalized.**
>
> Thank you for the reviewer’s feedback. To illustrate how the benchmark measures explainability, we refer to the example described in Fig. 3 of the paper. The query is: "_How should the aircraft configuration be adjusted to meet demand if the company decides to limit the number of Type A aircraft to no more than 15 and Type B aircraft to no more than 30?_"
>
> This example evaluates explainability in three ways:
> * Clear Decision Insights: The algorithm explains how constraints impact resource allocation, showing that the cost increases from $200,000 to $215,000 due to reduced flexibility.
> * Impact Analysis: The explanation quantifies trade-offs, demonstrating how the added constraints affect operational costs and resource usage.
> * Transparency: Detailed numerical results reveal how the constraints reshape the optimal solution, providing interpretable adjustments.
>
> While Fig. 3 outlines these explanations, future work will include interactive examples and demonstrations to further showcase the benchmark’s ability to evaluate and generalize explainability across various OR tasks.
>
> ### **W2: Explanation Quality Evaluation:**
>
> **Although the paper presents a dual evaluation (automated and expert) to assess explanation quality, it doesn’t fully explore how well these metrics align with user needs (who are, in other words, the major user groups if it is put into real-world use). For instance, it is unclear whether the explanation quality metrics are used to capture all the necessary aspects of understandability, particularly for non-experts. Providing more insights into how the explanations are judged, probably by including user feedback or more detailed breakdowns of the evaluation criteria, would offer a stronger case for the robustness of the framework.**
>
> Thank you for your thoughtful comments regarding the alignment of our explanation quality metrics with user needs, especially for non-expert users. Below is our response:
>
> **Coverage of User Needs:** Our framework evaluates explanations across four dimensions: result explanation, query relevance, causality, and sensitivity. These dimensions are carefully designed to meet the needs of both non-expert and expert users.
> * Non-expert users primarily rely on result explanation, which focuses on providing clear and interpretable outputs without requiring domain-specific knowledge.
> * Expert users benefit from additional dimensions such as query relevance, causality, and sensitivity, which provide deeper insights for complex decision-making.
>
> The evaluation process, including expert scoring, is structured around these dimensions, ensuring objectivity and consistency.
>
> **Ensuring Accessibility for Non-experts:** To address non-expert users’ needs, we prioritize clarity and accessibility in our evaluation framework. For example, automated metrics explicitly include criteria such as ease of understanding, as outlined in Appendix A.4. This ensures that the generated explanations are user-friendly and align with the expectations of diverse user groups.
>
> We plan to enhance the evaluation framework with a more detailed scoring rubric that assigns individual scores for clarity, completeness, and alignment with user needs. This will ensure a stronger balance between accessibility for non-experts and technical rigor for domain experts.

---

> ### Author Response · Authors · 2024-11-20
> **Thanks for your comment!**
>
> **W3: An analysis of computation complexity/scalability would also help strengthen the paper. I have potential concerns regarding the computational complexity of the proposed framework, especially given the use of LLMs and graph-based methods.**
>
> We appreciate your observation regarding the computational complexity of our proposed framework. In our framework, LLMs are used solely for generating explanations, making their contribution computationally lightweight. The response time is largely dependent on the choice of LLM. In our experiments, we utilized the GPT-4 series, which delivers real-time responsiveness with an average response time of approximately 3 seconds per query.
>
> During the explanation generation phase, we employ external tools such as bipartite graphs to quantify decision information. The edit distance computation for M constraints and N decision variables involves solving an NP-hard problem with a complexity of O((MN)!). While these methods require computational effort, their overhead is modest in our experiments. Empirical measurements indicate that this process adds approximately 10 seconds per query. Although not yet optimized for time complexity, this overhead is manageable within practical use cases.
>
> We recognize the importance of a more detailed computational complexity analysis to strengthen the scalability discussion. In future work, we plan to explore advanced graph-based algorithms to reduce time complexity and provide a rigorous theoretical and empirical evaluation of scalability.
>
> **Q1: Regarding the benchmark (for explainability): What were the criteria for selecting the problem categories, and how comprehensive is the benchmark in terms of covering different types of OR challenges? Additionally, have you validated the benchmark with industry professionals or in practical settings to ensure it reflects real-world complexity?**
>
> Thank you for your question. As mentioned in W1, the dataset was created using real-world, publicly available industry datasets, curated by experienced industry professionals. They selected problem categories and crafted queries based on practical business needs, focusing on diverse OR challenges like scheduling, routing, and resource allocation.
>
> The benchmark includes 300 high-quality queries designed to cover a wide range of OR problem types and complexities. To validate its representativeness, we used an iterative review and feedback process, comparing the queries to real-world business cases to ensure their applicability.
>
> We appreciate your feedback and will continue refining the benchmark in future iterations.
>
> **Q2: Regarding the system-level workflow: Have you considered how decision-makers would interact with the system and its explanations in real time? What if the system makes mistakes, can it be corrected under human supervision?**
>
> Thank you for your question. We have considered how decision-makers would interact with the system in real time. Our framework leverages AutoGen [2], which enables seamless integration of user feedback into the workflow. This design allows users to participate actively in the pipeline and provide inputs directly.
>
> If the system makes mistakes, our framework supports straightforward corrections through user interventions. For more complex corrections, targeted prompt designs or additional measures would be required. Nonetheless, the framework is highly extensible and adaptable to user-participation modes, ensuring decision-makers can interact effectively with the system.
>
> **Q3: Regarding the experimental setups/scalability: The paper compares EOR to OptiGuide, but have you considered comparing your work to other explainability frameworks outside the OR domain? Can this work be somehow extended to other similar domains but with different contexts, e.g., automated reasoning?**
>
> Thank you for your question. While our current comparison focuses on EOR and OptiGuide [3] within the OR domain, we acknowledge the value of comparing our work to explainability frameworks from other domains. This is something we plan to explore in future work to provide a broader perspective on EOR’s capabilities and performance.
>
> Regarding extensions to domains like automated reasoning, our framework's modular design and graph-based methods make it adaptable to similar contexts. While some domain-specific adjustments (e.g., tailored prompts or evaluation metrics) may be necessary, the core principles of our approach are flexible for domains requiring structured decision-making and explainability.
>
> References:
>
> [1] Tang, Z., et al. ORLM: Training Large Language Models for Optimization Modeling. arXiv preprint arXiv:2405.17743 (2024).
>
> [2] Wu, Q., et al. Autogen: Enabling next-gen llm applications via multi-agent conversation framework. arXiv preprint arXiv:2308.08155 (2023).
>
> [3] Li, B., et al. Large language models for supply chain optimization. arXiv preprint arXiv:2307.03875 (2023).

---

> > ### Comment · Reviewer_iUYd · 2024-11-20
> >
> > I appreciate the authors' detailed response, which has thoroughly addressed my concerns. If future revisions adequately incorporate these improvements, I will gladly consider raising my score.

---

> > > ### Author Response · Authors · 2024-11-22
> > > **Thanks for your feedback!**
> > >
> > > Thank you for your feedback. We have updated the paper with the suggested improvements and hope it meets your expectations. We sincerely appreciate your time and consideration.

---

> > > > ### Comment · Reviewer_iUYd · 2024-11-25
> > > >
> > > > I highly appreciate the authors' prompt revision and responses, which have addressed my concerns. Thus, I raise my score.

---

> > > > > ### Author Response · Authors · 2024-11-25
> > > > >
> > > > > Thank you so much for your positive feedback and for raising the score, we truly appreciate your great recognition of our work.
> > > > >
> > > > > Please don't hesitate to reach out if you have any further questions.

---

### Official Review · Reviewer_iVAa · 2024-11-01

**Soundness:** 2
**Presentation:** 2
**Contribution:** 2
**Rating:** 5
**Confidence:** 2

**Summary:**

The work introduces a framework called Explainable Operations Research (EOR) that aims at enhancing transparency in Operations Research (OR) models that incorporate Large Language Models (LLMs). The authors argue that, despite the prevalence of LLMs in OR, the lack of explanations represents a problem. EOR addresses this gap by introducing the concept of “decision Information”. This workflow involves three LLM agents to generate answers to user queries and corresponding explanations. The authors show that their method outperforms other methods in terms of quality of answers and explanations provided.

**Strengths:**

The paper makes a focused contribution by addressing a niche problem: explainability in OR. While this fits within the broader category of explainability for LLMs, the specificity of this context is important. The presentation is for the most part clear (but see below), and the agentic workflow proposed could be practically useful for practitioners.

**Weaknesses:**

Clarity and related work: The problem formulation would benefit from concrete examples to clarify what constitutes a "problem" and a "query" in the OR context. A more thorough description of OptiGuide would also help readers understand its role in the comparison. Bipartite graphs are mentioned multiple times, but it remains unclear why they are relevant or how they contribute to the framework—this needs clarification. The focus of EOR on coding tasks is not well explained, and it would be helpful to understand why this aspect is emphasized. Additionally, the paper proposes an agentic workflow but does not cite related literature; such citations should be included in the related work section.

Similarly, it’s also hard to judge the significance of the work because related work is hardly discussed. The workflow appears simple and I think that the novelty may not be enough for an ICLR paper.

Evaluation results: This is the main weakness. The benchmark used in experiments is not publicly available and is poorly described, making it hard to assess the quality of the experiments. The experimental section mentions a "proprietary LLM" without specifying which model is used, adding ambiguity to the evaluation. The explanation quality assessment relies on LLMs, but it is unclear if any manual evaluation by the authors was performed to validate these judgments. The experimental section needs many more results; currently, it's hard for the reviewer to judge the quality of the work.

**Questions:**

Posted as part of weaknesses.

---

> ### Author Response · Authors · 2024-11-20
> **Thanks for your comment!**
>
> Thank you for your thoughtful and constructive feedback. We appreciate the opportunity to address your concerns and clarify the contributions of our work.
>
> ### **Novelty and Significance:**
>
> **Practical Relevance and Research Gap**
>
> Our study addresses two critical gaps in OR. First, traditional "what-if" analyses predominantly focus on parameter changes, neglecting the impact of constraint variations, which are common in dynamic business environments. This oversight limits their practical applicability. We propose a novel framework that explicitly considers constraint changes, providing a more comprehensive approach to decision-making. Second, we develop a rigorous interpretability framework tailored to the large-model era, addressing the critical challenge of understanding and explaining the role of LLMs in OR contexts. By leveraging the publicly available industry dataset, IndustryOR [1], our work bridges a significant academic gap while addressing real-world industrial needs, offering valuable contributions to both theory and practice.
>
> **Key Contributions**
>
> * **Problem Definition:** We formulate the problem of explainable OR within the context of LLMs, establishing a foundation for future research in this emerging area.
> * **Introduction of Decision Information:** We propose the novel concept of decision information, utilizing bipartite graphs to quantify the importance of constraints in response to user queries. This approach enhances modeling capabilities and provides intuitive explanations when used in conjunction with LLMs, advancing the scope of what-if analysis in OR.
> * **Benchmark Development:** To evaluate explainability in OR, we develop a new benchmark that rigorously tests the effectiveness of explanations. This benchmark sets a new standard in the field and establishes metrics for future comparisons.
> * **Automation and Accessibility:** By leveraging LLMs, our framework automates key aspects of OR workflows, reducing dependence on various expert interventions and lowering communication costs. It also provides transparent and interpretable explanations, fostering greater trust and understanding among users.
>
> **Simplicity of Workflow**
>
> The streamlined design of our workflow is intentional and aimed at making complex OR techniques accessible to a broader audience without compromising analytical depth. This usability-focused approach bridges advanced methodologies and practical applications, ensuring relevance in diverse industrial scenarios. By presenting a straightforward yet powerful solution, we aim to lower the entry barrier for utilizing OR techniques and promote adoption in both academic and industrial settings.
>
> **Expanded Related Work Discussion**
>
> We recognize the importance of contextualizing our contributions within the broader landscape of existing literature. In our revision, we will include a more comprehensive review of related work to highlight how our approach enhances explainability in the field.
>
> We hope these clarifications address your concerns and demonstrate the novelty, significance, and practical value of our contributions. Thank you again for your constructive feedback, which will help us further refine our paper.

---

> ### Author Response · Authors · 2024-11-20
> **Thanks for your comment!**
>
> ### **Clarity and Related Work:**
>
> **The problem formulation would benefit from concrete examples to clarify what constitutes a "problem" and a "query" in the OR context.**
>
> Thank you for your feedback. We appreciate your suggestion to clarify the definitions of "problem" and "query" in the OR context with concrete examples.
>
> In our revised manuscript, we will introduce these clarifications earlier in the paper for improved readability. Specifically, A **problem** refers to a natural language description of the OR task to be solved. A **query** represents a hypothetical scenario or modification to the problem, such as "What if the delivery timeline is extended?" or "What if transportation costs increase by 15%?" For clarity, we will include a concrete example in the Introduction section, using a supply chain optimization scenario where the problem and associated queries are explicitly defined. We believe these adjustments will enhance understanding of our framework and its practical relevance.
>
> **A more thorough description of OptiGuide would also help readers understand its role in the comparison.**
>
> Thank you for your feedback. While we have already introduced OptiGuide [2] and its limitations in the Introduction as follows,
> _Another work, OptiGuide, emphasizes what-if analysis, which, while useful for specific easy scenarios, lacks the robust modeling capability to address more complex cases like deleting or combining constraints. For example, if a warehouse closes, the OR model must remove the related storage capacity constraint and adjust the distribution network. Current methods struggle to achieve this level of flexibility, yet such adaptability is crucial for accurately reflecting real-world changes. Most critically, the explanations these methods provide are often superficial, merely summarizing the outcomes without exploring the underlying reasons behind the results, thus lacking the quantitative analysis, depth, and clarity required to fully understand and trust the decision-making process._ We understand the need to provide additional details for better clarity. To address this:
>
> We will expand on OptiGuide’s functionalities and its emphasis on what-if analysis in a broader context, ensuring readers understand its primary contributions and limitations in handling complex scenarios like modifying or combining constraints. In the Baseline section, we will include a more detailed description of how OptiGuide was implemented and compared, including specific examples of its limitations.
>
> **Bipartite graphs are mentioned multiple times, but it remains unclear why they are relevant or how they contribute to the framework—this needs clarification.**
>
> Thank you for pointing this out. While the current version briefly mentions the role of bipartite graphs in the Introduction, we agree that a more detailed explanation is needed to clarify their relevance and contribution to our framework.
>
> To address this, we will expand the Introduction to explicitly explain how bipartite graphs are used to measure the importance of decision factors in response to user queries, particularly under scenarios involving complex changes in constraints.
>
> In Section 3.2.2, we will provide a more comprehensive discussion, including the specific mechanisms through which bipartite graphs contribute to our framework and examples that illustrate their impact on decision-making.
>
> These updates will ensure that the role of bipartite graphs is clear and well-integrated into the overall framework.
>
> **The focus of EOR on coding tasks is not well explained, and it would be helpful to understand why this aspect is emphasized.**
>
> Thank you for your comment. We recognize the need to explain the emphasis on coding tasks better in the context of EOR. As shown in Figure 1, solving OR problems typically involves two main steps: first, modeling the problem, and second, converting the model into executable code to call external solvers for solutions. Coding, therefore, plays a critical role in bridging the problem modeling and solution process.
>
> To address this feedback, we will revise the Introduction to explicitly highlight the importance of coding in OR problem-solving, ensuring readers understand why this aspect is central to our framework.
>
> **Additionally, the paper proposes an agentic workflow but does not cite related literature; such citations should be included in the related work section.**
>
> Thank you for your feedback. In our initial submission, we focused primarily on reviewing literature at the intersection of OR and LLMs. We recognize the importance of situating our proposed agentic workflow within the broader context of agent frameworks. To address this, we will expand the Related Work section to include citations from more general research on agent frameworks, ensuring a more comprehensive contextualization of our approach.

---

> ### Author Response · Authors · 2024-11-20
> **Thanks for your comment!**
>
> ### **Evaluation results:**
>
> **The benchmark used in experiments is not publicly available and is poorly described, making it hard to assess the quality of the experiments.**
>
> Thank you for highlighting this issue. In the current version, we recognize that the description of the benchmark dataset is insufficiently detailed. To address this:
>
> We will include a more comprehensive explanation in the revised version, detailing the process by which the dataset was constructed. Specifically, we will describe how three OR experts with PhDs and significant industry experience designed 10 unique queries for each problem, ensuring the queries reflect realistic business scenarios. These scenarios include variations in parameters (e.g., cost, resource limits) and constraints (e.g., adding/removing time windows or resource allocations).
>
> We will further elaborate on the dataset’s scope and diversity, including its iterative review process and alignment with real-world business cases.
>
> Furthermore, we commit to making the benchmark publicly available upon the publication of the paper.
>
> **The experimental section mentions a "proprietary LLM" without specifying which model is used, adding ambiguity to the evaluation.**
>
> Thank you for pointing this out. In the current version, we used the term "proprietary LLM" as a general descriptor when discussing the baseline. To clarify, all methods in our experiments were evaluated using the same LLM model to ensure a fair comparison.
>
> In the revised version, we will explicitly state the specific LLM model used (such as, GPT-4-Turbo) in the experiments and emphasize that the same model was consistently applied across all methods. This update will eliminate any ambiguity and ensure transparency in the evaluation process.
>
> **The explanation quality assessment relies on LLMs, but it is unclear if any manual evaluation by the authors was performed to validate these judgments.**
>
> We acknowledge the need to clarify how explanation quality was assessed in the experiments. We understand that manual evaluation by human experts is a gold standard for assessing explanation quality. In fact, we performed a manual evaluation through the Expert method, as described in the paper, where human experts with extensive domain knowledge reviewed the explanation quality to ensure a thorough and reliable assessment.
>
> However, we also recognize that manual evaluation can be time-consuming and resource-intensive, making it impractical for large-scale or real-time applications. To address this challenge, we proposed the Auto method, which utilizes LLMs for automated assessment of explanation quality. Our experiments demonstrate that the results from the Auto method are highly consistent with those of the Expert method, suggesting that LLM-based evaluations can serve as a practical and reliable alternative.
>
> **The experimental section needs many more results; currently, it's hard for the reviewer to judge the quality of the work.**
>
> In our paper, we have reported experimental results on multiple aspects, including:
>
> * Modeling Accuracy: Detailed evaluations of the accuracy of the generated solutions.
> * Explanation Quality: Assessment of explanation quality through both manual evaluation by experts and an automated evaluation method.
> * Case Studies: Analysis of specific examples that demonstrate the quality of our generated explanations and their application in realistic scenarios.
> * Failure Cases: Examination of scenarios where our approach did not perform as expected, with a discussion on potential improvements to address these limitations.
>
> In response to Reviewer ko7t’s feedback, we have also included parameter sensitivity analysis to evaluate the reliability and stability of our approach under varying conditions.
>
> We would greatly appreciate your suggestions on whether there are additional aspects or types of results that you feel could further strengthen the experimental section. We are committed to improving the quality of the work.
>
> References:
>
> [1] Tang, Z., et al. ORLM: Training Large Language Models for Optimization Modeling. arXiv preprint arXiv:2405.17743 (2024).
>
> [2] Li, B., et al. Large language models for supply chain optimization. arXiv preprint arXiv:2307.03875 (2023).

---

> ### Author Response · Authors · 2024-12-02
> **Humbly Awaiting Your Feedback on Rebuttal for 8688.**
>
> Dear Reviewer iVAa,
>
> We thank you again for your time reviewing our paper! We have dealt with all your concerns to the best of our ability and carefully incorporated your valuable suggestions into the updated manuscript. As the end of the author-review discussion period is approaching, we would appreciate it if you could kindly let us know whether your previous concerns have been fully addressed. We are more than happy to answer any further questions you might have.
>
> As you also mentioned that our work conveys sufficiently novel message and insight for explainable OR within LLMs, if you think our work is worthy of a higher score, we would be immensely grateful!
>
> Thank you again for your time, effort, and valuable feedback. We look forward to your kind response.

---

### Official Review · Reviewer_1vGE · 2024-11-03

**Soundness:** 3
**Presentation:** 2
**Contribution:** 3
**Rating:** 6
**Confidence:** 3

**Summary:**

This paper proposes a LLM-based explainable operations research (EOR) framework and evaluates it on a benchmark they created. EOR is a multi-agent framework that starts with a user query and generates an OR program (for Gurobi optimization solver), uses the program to solve the problem and generates explanations of the code as well as answers using LLMs. The experimental evaluation of accuracy and explanations shows that EOR outperforms two baselines.

**Strengths:**

1. The paper proposes a novel use of LLMs for explainability in the field of Operations Research. The experiments show that such approach is effective as per the ratings given by LLMs as well as human OR experts.
2. Example 3 is quite useful in understanding the task as well as the output of EOR.

**Weaknesses:**

1. While the paper reads well, some parts are not very clear. In particular:
 - It is not clear how the decision information, the bi-partite graph, the edit distance computation etc. is connected to LLMs. What does `Since LLMs cannot directly perform this quantification, we utilize them to sense these processes and generate explanatory insights.` exactly translate to in figure 2?
- The section 3.1 on problem formulation focusses on explanations. What is the approach/innovation for modeling that translates to improved modeling accuracy in Table 2?
2. The dataset is not shared, the paper states `Notably, the question sets and the queries in this benchmark are developed from scratch and managed in-house`. Would it be made available publicly? Does `developed from scratch` mean manual creation?
3. If EOR enhances the modeling process, could you compare modeling accuracy on other public benchmarks?

**Questions:**

1. Are the ground truth expert ratings on the quality of explanations part of the benchmark?
2. Why is Table 3 missing some cells?

---

> ### Author Response · Authors · 2024-11-20
> **Thanks for your comment!**
>
> We sincerely appreciate the constructive suggestions and comments on our work. The following are our detailed responses to each of the reviewer's comments.
>
> **W1-1: It is not clear how the decision information, the bi-partite graph, the edit distance computation etc. is connected to LLMs. What does `Since LLMs cannot directly perform this quantification, we utilize them to sense these processes and generate explanatory insights.` exactly translate to in figure 2?**
>
> Decision Information is a concept we introduce to extend traditional what-if analysis, which focuses on parameter changes (sensitivity analysis), by enabling the analysis of constraint changes. The bipartite graph transformation and edit distance computation quantify the impact of query-induced model changes, which is how we measure Decision Information.
>
> Since LLMs cannot directly perform this quantification, we employ an external tool to handle the decision information analysis. This computed information is then provided to the LLM as auxiliary input, enabling it to incorporate the decision information process into the generated explanations. This ensures that the LLM's outputs are informed by the quantified impact of the changes.
>
> In Fig. 2, which provides a high-level overview of our pipeline, steps (6), (7), and (8) represent the processes related to these concepts. While the figure abstracts away the implementation details, these steps include the integration of decision information into the LLM's explanation generation process. We will revise the figure to better highlight these connections and clarify their role.
>
> **W1-2: The section 3.1 on problem formulation focusses on explanations. What is the approach/innovation for modeling that translates to improved modeling accuracy in Table 2?**
>
> First, explanations are meaningful only when based on accurate modeling results. If the modeling is incorrect, the solution will also be incorrect, rendering the explanation meaningless. The modeling accuracy reported in Table 2 establishes a solid foundation for generating valid and trustworthy explanations.
>
> Second, modeling accuracy indirectly reflects the quality of our explanations. Specifically, our approach includes Explanation of Correctness, which evaluates the generated code. Accurate modeling results validate the correctness of the generated code, thereby reinforcing the reliability of our explanations.
>
> **W2: The dataset is not shared, the paper states `Notably, the question sets and the queries in this benchmark are developed from scratch and managed in-house.` Would it be made available publicly? Does developed from scratch mean manual creation?**
>
> Yes, we plan to make our dataset publicly available upon the acceptance of the paper. To ensure high quality and practical relevance, our dataset was manually created by three OR experts with PhDs and over three years of industry experience.
>
> Additionally, because the dataset was developed entirely in-house, it guarantees that the queries have not been seen by existing LLMs, eliminating the risk of data leakage.
>
> **W3: If EOR enhances the modeling process, could you compare modeling accuracy on other public benchmarks?**
>
> Since modeling is an integral part of our framework, it is indeed possible to evaluate our modeling accuracy on existing public datasets. However, the primary focus of our work is on explanations, and existing public datasets do not provide the features necessary to evaluate the quality of explanations.
>
> For this reason, we developed a custom dataset tailored for explanation-focused evaluations from the open-source IndustryOR [1], addressing both modeling and explanation needs. We appreciate your feedback and will explore further comparisons if relevant public datasets become available.
>
> **Q1: Are the ground truth expert ratings on the quality of explanations part of the benchmark?**
>
> The expert ratings correspond only to the specific explanations generated in this study and serve as a quality metric rather than a part of the benchmark itself. Our benchmark includes only the ground truth for correct results, not expert-generated explanations. While incorporating expert-generated explanations might seem ideal, it presents significant challenges. The correct results remain fixed and consistent across experts, but the logic and language used in their explanations can vary significantly due to inherent subjectivity. This variability makes it difficult to establish a standardized benchmark. To our knowledge, no prior work has adopted such an approach.
>
> **Q2: Why is Table 3 missing some cells?**
>
> The missing cells in Table 3 result from baseline methods not generating explanations for the generated code, leaving no expert ratings available for these methods.
>
> References:
>
> [1] Tang, Z., et al. ORLM: Training Large Language Models for Optimization Modeling. arXiv preprint arXiv:2405.17743 (2024).

---

> > ### Comment · Reviewer_1vGE · 2024-11-25
> >
> > I thank the authors for their detailed responses.
> > I have some follow-up questions:
> >
> > * W1-2: In figure 2, there is no arrow from explanations to commander indicating that the explanations should not have any effect on modeling code or answers. That still does not explain the modeling improvements noticed in Table 2.
> >
> > * The response to W3 states that `developed a custom dataset tailored for explanation-focused evaluations`, but the response to Q1 is that `Our benchmark includes only the ground truth for correct results, not expert-generated explanations`. Could you please clarify how you imagine others can use your dataset for explanation evaluations? Also, even though expert generated ground truth could be subjective, it is accepted for semantic-similarity based metrics. In NLP, one such example is summarization, the ground truth for summarization is not objective.

---

> ### Author Response · Authors · 2024-11-26
> **Thanks for your feedback!**
>
> Thank you for your feedback. We have carefully reviewed your new comments and provided further clarifications in the hope of resolving your concerns.
>
>
> ### **For the first question**
>
> * **Clarification on Explanations:** The explanations in our framework are not designed to influence the modeling process, including code generation or accuracy. Instead, their purpose is to support better decision-making. These explanations are generated based on accurate modeling results, ensuring that they are meaningful only when the underlying modeling is correct.
> * **Regarding Table 2 Results:** The improvements shown in Table 2 reflect the accuracy and performance of the modeling component within our framework, which operates independently of the explanation part. While explanations are derived from the correctly generated code, they do not guide or enhance the modeling process itself. Instead, accurate modeling results confirm the correctness of the generated code, which in turn supports the reliability and credibility of the explanations.
>
> In summary, explanations depend on accurate modeling but do not influence the modeling process. The performance improvements shown in Table 2 arise from the modeling ability of our framework, not from the explanations' influence.
>
>
> ### **For the second question**
>
> Our explainable OR framework targets classical OR scenarios, especially what-if analysis, where user queries are essential inputs. We mainly focus on explaining why new solutions are generated. For example, in the query from our paper, _How should the aircraft configuration be adjusted if the company limits Type A aircraft to 15 and Type B aircraft to 30?_, the goal is to understand how this new query impacts the OR results, thereby supporting better decision-making.
>
> As noted in the response to the first question, accurate modeling is foundational because explanations rely on correct modeling results. However, explanations remain our primary focus. Existing OR benchmarks typically include natural language descriptions and correct answers for the OR problems but lack user queries. Since our framework requires user queries as inputs, it cannot be evaluated on these datasets, which focus solely on modeling without considering user queries.
>
> We recognize that high-quality, expert-generated explanations as benchmarks, along with semantic-similarity metrics, would be ideal for evaluating explanation quality, as seen in NLP tasks like summarization. However, in OR tasks, generating such explanations is highly impractical due to the significant costs involved. These costs include the need for experienced experts, the significant time investment required, and the effort necessary to ensure cross-checked quality, all of which make standardization challenging.
>
> Our approach addresses these challenges at its core by providing a reliable and scalable method for evaluating explanation quality. This method is built upon two key elements:
>
> * **Objective Ground Truth:** OR results are definitive, objective, and critical. Explanation quality is irrelevant without correct results. Providing correct solutions ensures objectivity and consistency, avoiding subjective variations in logic or language by humans. Therefore, we use the correct result for each query as the ground truth.
> * **Reliable Evaluation Methods:** We employ the balanced and reliable approach to evaluate explanation quality. This includes low-cost expert ratings for explanation quality and an automated evaluation method as a scalable alternative, validated for consistency with expert scores. Specifically, our framework evaluates explanations across four dimensions of most significant concern to the OR field: result explanation, query relevance, causality, and sensitivity. These dimensions are carefully designed to meet the needs of both non-expert and expert users. This dual approach ensures both objectivity and practical evaluation of explanations.
>
> We sincerely hope these clarifications address your concerns, and we deeply appreciate the opportunity to provide further explanations. Please do not hesitate to reach out if you have any additional questions.

---

> > ### Comment · Reviewer_1vGE · 2024-11-26
> >
> > * Right, so my original question was `What is the approach/innovation for modeling that translates to improved modeling accuracy in Table 2?`
> >
> > * For the statement `existing public datasets do not provide the features necessary to evaluate the quality of explanations`, could you elaborate on what features are necessary and confirm which of those features does IndustryOR have?

---

> > > ### Author Response · Authors · 2024-11-26
> > > **Thanks for your prompt feedback!**
> > >
> > > Thank you for your prompt feedback. We have carefully reviewed your latest comments and provided additional clarifications with the sincere hope of addressing your concerns.
> > >
> > >
> > > **NQ1: What is the approach/innovation for modeling that translates to improved modeling accuracy in Table 2?**
> > >
> > > As shown in Figure 2 of the paper, we propose a **multi-agent framework** to improve modeling accuracy. Specifically, for steps (1), (2), (3), (4), and (5) in the figure, we enhance accuracy through the coordinated actions of multiple agents. When a user submits a query, a **Commander** agent coordinates the **Writer** and **Safeguard** agents. The Writer analyzes the query with carefully crafted prompts to determine whether to add, delete, or update the original code for the OR problem. If the generated code contains errors, the Safeguard agent assists by debugging and ensuring correctness. For more details, please refer to Section 3.2.1 of the paper and Appendix A.2, which provides the prompt template design.
> > >
> > >
> > > **NQ2: Could you elaborate on what features are necessary and confirm which of those features does IndustryOR have?**
> > >
> > > Thank you for your follow-up question. As mentioned, our focus is on classic OR scenarios like **what-if analysis**, which are crucial for supporting better decision-making. We aim to develop an open-source industry benchmark for **explainable OR** within LLMs.
> > >
> > > To achieve this goal, the benchmark must include the following features:
> > > * **User queries paired with the correct answers**, which are essential for both benchmark and explanations (as previously explained). Unfortunately, existing open-source datasets, including IndustryOR, do not currently include such query-answer pairs. They are primarily designed for modeling purposes and are not suited for decision-making explanations based on what-if analysis.
> > > * **High-quality OR problems** that closely reflect real-world application scenarios. On this front, the IndustryOR dataset meets the criteria with its well-aligned, practical scenarios. This is precisely why we derived our problems from IndustryOR, rather than other publicly available datasets.
> > >
> > > We sincerely appreciate your thoughtful review and hope this explanation clarifies the necessary features and how IndustryOR aligns with them.
> > >
> > >
> > > We deeply value the opportunity to address your concerns and provide further explanations. Please feel free to reach out if you have any additional questions or require further details.

---

> > > > ### Comment · Reviewer_1vGE · 2024-11-27
> > > >
> > > > Thank you for the explanations. Based on my improved understanding, I have raised the score.

---

> > > > > ### Author Response · Authors · 2024-11-27
> > > > > **Thanks for your feedback!**
> > > > >
> > > > > Thank you so much for your positive feedback and for raising the score, we truly appreciate your recognition of our work.
> > > > >
> > > > > Please don't hesitate to reach out if you have any further questions.

---

### Official Review · Reviewer_ko7t · 2024-11-04

**Soundness:** 3
**Presentation:** 3
**Contribution:** 3
**Rating:** 8
**Confidence:** 4

**Summary:**

The authors introduce a framework emphasizing actionable and understandable operational research explanations. They explore how constraints affect decision-making and introduce an industrial benchmark to assess the effectiveness of explanations in operational research.

**Strengths:**

- The authors tackle a relevant real-world problem
- The authors explore the intersection of explainability for operational research and LLMs - a hot topic right now

**Weaknesses:**

- the authors compare their method against only one baseline
 - while one of the key contributions is the development of a benchmark dataset, it is not clear how the data was gathered or what tasks and metrics are considered for it. While some are mentioned, they are not described in detail.
 - the authors mention the concept of decision information as one of the pillars of explainable operational research. Nevertheless, it is unclear whether results were reported for it.
 - while the authors claim that their framework enables actionable and understandable explanations, they only assess the quality of explanations considering attribution and justification explanations, with no reference to evaluating whether and how actionable the explanations are.

**Questions:**

Dear authors,

We consider the research relevant and interesting. Nevertheless, we would like to point to some improvement opportunities:

GENERAL COMMENTS

(1) - We encourage the authors to strengthen the related work considering works related to explainability for operations research. E.g., (a) De Bock, Koen W., et al. "Explainable AI for operational research: A defining framework, methods, applications, and a research agenda." European Journal of Operational Research 317.2 (2024): 249-272. and (b) Thuy, Arthur, and Dries F. Benoit. "Explainability through uncertainty: Trustworthy decision-making with neural networks." European Journal of Operational Research 317.2 (2024): 330-340. Furthermore, the authors could briefly reference explainability metrics mentioned in the literature relevant to this work.

(2) - The authors compare the proposed method only against the OptiGuide baseline. Are there any additional baseline methods they could compare to?

(3) - Do the results reported for Auto vs. Expert only consider the cases when the LLM did not display any kind of errors? Do the scores degrade if considering cases with errors vs. the ones displayed by the baseline model?

(4) - In Section 4.7, the authors provide detailed insight into the failure cases for the proposed method. What are the failure ratios for the baseline method and error typification? Is the proposed method more reliable in this regard? Could the authors increase the number of executions? For example, having two failures in 14 executions is different from observing the same proportion in 100 runs.

(5) - One of the key contributions of the paper is the creation of a new industrial benchmark dataset specifically tailored to assess the effectiveness of explanations in OR. We would invite the authors to introduce it in greater detail: (a) how was the data obtained?, (b) what tasks were considered and why?, (c) how were the queries created/what experience informed the queries creation and how it was validated that they capture a wide range of scenarios?, (d) what metrics are considered in the benchmark and what is their underlying rationale?, (e) are there any other benchmarks in the field or close domains that should be considered (e.g., to draw inspiration)?, (f) is the benchmark published/will be published?

(6)- The authors mention two aspects being measured to assess the quality of explanations: attribution and justification. We would appreciate some examples of how such explanations look and how they are assessed for correctness.

(7)—One of the framework's key contributions is the generation of actionable explanations. Nevertheless, we miss a thorough evaluation of this aspect in the explanations generated. Furthermore, it is unclear to us whether this aspect was also included in the proposed benchmark.

(8) - One of the paper's key contributions is the concept of Decision Information and its quantification. Nevertheless, no results have been reported on it. Do the authors consider that reporting it could enhance the understanding of their research work and provide an additional perspective to the ones already exposed in the manuscript?

FIGURES

(9) - Figure 3: We encourage the authors to provide a brief explanation of what the green/yellow highlighted explanations mean. Do they refer to attribution and justification explanation aspects?

TABLES

(10) - Table 2: in the caption, please indicate what does bolding the results mean

(11) - Table 3: highlight the best results (e.g., by bolding them)

(12) - All tables reporting metrics: provide arrows next to the metric names (up/down) indicating whether a higher/lower result is better.

---

> ### Author Response · Authors · 2024-11-20
> **Thanks for your comment!**
>
> We sincerely appreciate the constructive suggestions and comments on our work, which have definitely helped us to enhance the paper. The following are our detailed responses to each of the reviewer's comments.
>
> **Q1: We encourage the authors to strengthen the related work considering works related to explainability for operations research. ... Furthermore, the authors could briefly reference explainability metrics mentioned in the literature relevant to this work.**
>
> Thanks to the reviewer for the helpful suggestions on related work. While our paper primarily focuses on explaining solving OR problems within LLMs, the references [1, 2] provided offer valuable insights into explainability developments for non-LLMs in OR. We reviewed these works in detail and will add them to our related work section. Additionally, we will cite relevant explainability metrics and methods from these sources, such as accuracy metrics and expert evaluation methods, to deepen our study's context.
>
> **Q2: The authors compare the proposed method only against the OptiGuide baseline. Are there any additional baseline methods they could compare to?**
>
> Currently, OptiGuide [3] is the only available and relevant baseline for this specific problem, and no other established baseline methods exist for direct comparison in the literature.
>
> **Q3: Do the results reported for Auto vs. Expert only consider the cases when the LLM did not display any kind of errors? Do the scores degrade if considering cases with errors vs. the ones displayed by the baseline model?**
>
> Yes, we only consider cases where the model results are correct when comparing the explanation quality between Auto and Expert. As emphasized in the paper Sec. 4.5.2, if the model produces the wrong solution or errors, interpreting those results is not meaningful, so cases with modeling errors are excluded from our evaluation.
>
> **Q4: In Section 4.7, the authors provide detailed insight into the failure cases for the proposed method. What are the failure ratios for the baseline method and error typification? Is the proposed method more reliable in this regard? Could the authors increase the number of executions? For example, having two failures in 14 executions is different from observing the same proportion in 100 runs.**
>
> Our analysis of failure cases primarily aims to understand the strengths and weaknesses of our proposed model, identifying areas where it is prone to errors to inform targeted future improvements. Therefore, we did not analyze failure cases for the baseline method.
>
> Temperature is a parameter that controls the “creativity” or randomness of the text generated by LLMs such as GPT-4. A higher temperature (e.g., more than 1) results in more diverse and creative output, while a lower temperature (e.g., 0) makes the output more deterministic and focused, ideal for tasks requiring factual accuracy such as OR modeling. To ensure reproducibility, we set the LLM temperature to 0, as commonly done, making modeling accuracy depend solely on the number of debugging attempts. Following the settings in [3], we initially set the debug attempts to 3. Based on the reviewer’s suggestion, we conducted additional experiments: 1) Tested temperatures of 0.5 and 1 to assess result reliability. 2) Increased debugging attempts to 10 to evaluate performance stability.
>
> | Setting    | Temperature | OptiGuide | EOR      |
> |------------|-------------|-----------|----------|
> | Zero-shot  | 0           | 30.33%    | 88.33%   |
> |            | 0.5         | 22.00%    | 89.67%   |
> |            | 1           | 26.33%    | 88.00%   |
> | One-shot   | 0           | 69.33%    | 95.33%   |
> |            | 0.5         | 70.00%    | 93.67%   |
> |            | 1           | 68.33%    | 93.33%   |
>
> | Setting    | Debug Times | GPT-4-1106-preview | GPT-4-Turbo |
> |------------|-------------|---------------------|-------------|
> | Zero-shot  | 3           | 81.67%              | 88.33%      |
> |            | 10          | 81.33%              | 89.00%      |
> | One-shot   | 3           | 87.67%              | 95.33%      |
> |            | 10          | 87.67%              | 95.33%      |
>
> From these two tables' results, we can find that:
>
> * Temperature Variation: The results demonstrate that the model's performance remains stable across different temperature settings (0, 0.5, 1) in both zero-shot and one-shot scenarios.
> * Debug Attempts: Increasing the number of debug attempts from 3 to 10 does not improve performance, indicating that more attempts mainly lead to unnecessary resource usage without enhancing results.
>
> We will update these results in the new version, including details in the Appendix.

---

> ### Author Response · Authors · 2024-11-20
> **Thanks for your comment!**
>
> **Q5: We would invite the authors to introduce it in greater detail: (a) how was the data obtained?, (b) what tasks were considered and why?, (c) how were the queries created/what experience informed the queries creation and how it was validated that they capture a wide range of scenarios?, (d) what metrics are considered in the benchmark and what is their underlying rationale?, (e) are there any other benchmarks in the field or close domains that should be considered (e.g., to draw inspiration)?, (f) is the benchmark published/will be published?**
>
> After examining publicly available OR datasets, we identified a high-quality open-source dataset, IndustryOR [4], which partially aligned with our needs. However, it lacked the interpretability features we required. Therefore, we refined this dataset by selecting 30 diverse problems from its 3,000 entries (these entries contain many duplicate or similar data). These problems cover a broad range of real-world business scenarios, such as supply chain management, financial investment, airline scheduling, and logistics management, providing a representative cross-section of OR applications.
>
> To ensure the high quality and practical relevance of the queries, we invited three OR experts with PhDs and more than three years of industry experience to design 10 unique queries for each problem. Leveraging their extensive experience, these experts created queries that reflect realistic business conditions, introducing variations such as parameter adjustments (e.g., cost, resource limits, or priorities) and modifications to constraints (e.g., adding or removing time windows or resource allocation restrictions). This process ensures that the queries cover diverse and realistic business scenarios.
>
> The final dataset comprises 300 queries with a wide-ranging scope, diversity, and high quality. To validate the queries, we implemented an iterative review and feedback mechanism, which included comparisons to real-world business cases to ensure their representativeness and applicability.
>
> To evaluate the benchmark, we adopted a dual approach. Similar to current modeling methods, we assess performance based on accuracy-related metrics. The reason why we need an accuracy metric is that an explanation is meaningless if the solution is wrong. Additionally, given the importance of evaluating the generated explanations, we developed an automated method for assessing explanation quality. While the ideal evaluation method would involve human experts scoring the explanations, such an approach is impractical due to the significant time, effort, and costs involved. Instead, we designed an automated evaluation system and validated its reliability by comparing its results against human expert assessments. This ensures that the automated evaluation provides a reasonable proxy for expert judgment, making it efficient yet credible.
>
> While OptiGuide [3] shares some similarities, it focuses on supply chain scenarios and simpler query transformations, with limited scope and no open-source access. Our benchmark, covering a broader range of scenarios with greater complexity, will be made publicly available upon publication to support further research.
>
> We introduced the benchmark in Sec. 4.1; however, we acknowledge that it is not yet complete. Based on the reviewer's feedback, we will be adding more details and refinements to provide greater clarity and comprehensiveness.
>
> **Q6: The authors mention two aspects being measured to assess the quality of explanations: attribution and justification. We would appreciate some examples of how such explanations look and how they are assessed for correctness.**
>
> Yes, our study categorizes LLM-generated explanations for OR problems into two main types: attribution explanations and justification explanations. However, the primary focus of our paper is on justification explanations.
>
> For justification explanations, we assess them from two perspectives:
>
> * Explanation of Correctness: This evaluates the logical consistency and validity of the generated code in relation to the solution. To measure the correctness of the explanation, we use accuracy as an indirect metric to assess its effectiveness.
> * Explanation of the Results: This focuses on the interpretive quality of the generated textual explanations, leveraging both automated metrics and expert evaluations to provide a comprehensive assessment of explanation quality.
>
> Examples of what justification explanations look like can be found in Fig. 3 of our paper. This figure provides detailed illustrations of the structure and content of these explanations. For both types of explanations, we employ a dual evaluation approach combining expert reviews and automated methods to ensure a robust and reliable assessment.

---

> ### Author Response · Authors · 2024-11-20
> **Thanks for your comment!**
>
> **Q7: One of the framework's key contributions is the generation of actionable explanations. Nevertheless, we miss a thorough evaluation of this aspect in the explanations generated. Furthermore, it is unclear to us whether this aspect was also included in the proposed benchmark.**
>
> Our framework emphasizes evaluating textual explanations generated by LLMs, focusing on their interpretive alignment with OR solutions. As addressed in Q6, we assess explanations from two perspectives: 1) Accuracy: An indirect measure of explanation correctness. 2) Quality: A comprehensive evaluation using automated metrics and expert reviews.
>
> These evaluations are not part of the benchmark. While incorporating expert-generated explanations into the benchmark is ideal, it poses challenges: expert biases, diverse explanation styles, and subjective formats hinder standardization. To our knowledge, no prior work has implemented such an approach.
>
> Instead, our benchmark includes the true solution, as correctness is paramount: 1) Explanation quality is irrelevant without correct results. 2) Providing solutions ensure objectivity and consistency, avoiding subjective variations in logic or language by humans.
>
> We hope this clarifies our rationale. Thank you for your valuable feedback!
>
> **Q8: One of the paper's key contributions is the concept of Decision Information and its quantification. Nevertheless, no results have been reported on it. Do the authors consider that reporting it could enhance the understanding of their research work and provide an additional perspective to the ones already exposed in the manuscript?**
>
> The concept of Decision Information in our work extends traditional what-if analysis by not only addressing parameter changes (sensitivity analysis) but also incorporating changes in constraints. We quantify Decision Information by transforming the generated models into bipartite graphs and calculating the edit distance between them. This quantification is then integrated into the LLM’s input, enabling the generation of explanations that reflect this perspective.
>
> We believe the current manuscript demonstrates this process effectively, as the quantified Decision Information directly influences the explanations presented in our results. However, we appreciate your feedback and will consider including additional data or examples in future revisions to further enhance clarity and provide deeper insights.
>
> **Q9: Figure 3: We encourage the authors to provide a brief explanation of what the green/yellow highlighted explanations mean. Do they refer to attribution and justification explanation aspects?**
>
> Thank you for your suggestion. You are correct: the green and yellow highlights in Fig. 3 represent attribution and justification explanation aspects. We will revise the paper to clarify this for better reader understanding.
>
> ### Tables
>
> **Q10: Table 2: in the caption, please indicate what does bolding the results mean**
>
> **Q11: Table 3: highlight the best results (e.g., by bolding them)**
>
> **Q12: All tables reporting metrics: provide arrows next to the metric names (up/down) indicating whether a higher/lower result is better.**
>
> Thank you for your suggestions. We will revise the tables accordingly by clarifying bolded results, highlighting the best values, and adding arrows to indicate whether higher or lower metrics are better.
>
> References:
>
> [1] De Bock, K. W. et al. Explainable AI for Operational Research: A defining framework, methods, applications, and a research agenda. European Journal of Operational Research 317, 249–272 (2024).
>
> [2] Thuy, A. & Benoit, D. F. Explainability through uncertainty: Trustworthy decision-making with neural networks. European Journal of Operational Research 317, 330–340 (2024).
>
> [3] Li, B., et al. Large language models for supply chain optimization. arXiv preprint arXiv:2307.03875 (2023).
>
> [4] Tang, Z., et al. ORLM: Training Large Language Models for Optimization Modeling. arXiv preprint arXiv:2405.17743 (2024).

---

> > ### Comment · Reviewer_ko7t · 2024-11-26
> >
> > Dear authors,
> >
> > I have read the responses, and I will update the score and improve it a bit. Thank you for the detailed responses!

---

> > > ### Author Response · Authors · 2024-11-27
> > > **Thanks for your feedback!**
> > >
> > > Thank you so much for your positive feedback and for raising the score, we truly appreciate your great recognition of our work.
> > >
> > > Please don't hesitate to reach out if you have any further questions.

---

### Author Response · Authors · 2024-11-22
**For all reviewers: paper revision**

Dear Reviewers,

Thank you for your valuable feedback and constructive suggestions. We have carefully considered your comments and made the following key revisions to our paper. The updated version has now been uploaded for your review.

* **Enhanced Benchmark Description:** We have updated the description of our benchmark to emphasize its data sources, construction process, and relevance to real-world cases.
* **Improved Presentation and Clarity:** The presentation of our tables has been refined, and the introduction section now includes brief explanations of key concepts (such as query) to enhance reader understanding. Additionally, minor clarifications, such as the potential inclusion of user feedback and the meaning of colors (e.g., green and yellow), have been addressed.
* **Expanded Literature Review:** We have incorporated additional related work to provide a more comprehensive context for our research.
* **Hyperparameter Sensitivity Analysis:** A detailed experiment on hyperparameter sensitivity has been added in Appendix A.4.

We hope these revisions effectively address your concerns, and we are deeply grateful for your insights, which have significantly improved our work.

---

### Author Response · Authors · 2024-11-25

Dear Reviewers,

We thank you again for your time reviewing our paper! We have dealt with all your concerns to the best of our ability and carefully incorporated your valuable suggestions into the updated manuscript. As the end of the author-review discussion period is approaching, we would appreciate it if you could kindly let us know whether your previous concerns have been fully addressed. We are more than happy to answer any further questions you might have.

As you also mentioned that our work conveys sufficiently novel message and insight for explainable OR within LLMs, if you think our work is worthy of a higher score, we would be immensely grateful!

---

### Meta-Review · Area_Chair_2Gsh · 2024-12-16

**Metareview:**

After reading the reviewers' comments, and reviewing the paper, we recommend acceptance for Poster.

Below a detailed description of the paper, with key strengths and possibly remaining weaknesses.

The paper proposes a novel LLM-based explainable operations research framework that is evaluate on the a benchmark created by the authors. The experimental evaluation show that
their framework outperforms two baselines in terms of accuracy and explanations.

The key strengths (S#) of the paper are as follows:

- (S1)	The authors tackle a relevant real-world problem
- (S2)	The paper proposes a novel use of LLMs for explainability in the field of Operations Research. The experiments show that such approach is effective as per the ratings given by LLMs as well as human OR experts.
- (S3)	The paper makes a focused contribution by addressing a niche problem: explainability in OR. While this fits within the broader category of explainability for LLMs, the specificity of this context is important. The presentation is for the most part clear (but see below), and the agentic workflow proposed could be practically useful for practitioners.

The key weaknesses (W#) are as follows:

- (W1)	Clarity and related work: The problem formulation would benefit from concrete examples to clarify what constitutes a "problem" and a "query" in the OR context. A more thorough description of OptiGuide would also help readers understand its role in the comparison. Bipartite graphs are mentioned multiple times, but it remains unclear why they are relevant or how they contribute to the framework—this needs clarification. The focus of EOR on coding tasks is not well explained, and it would be helpful to understand why this aspect is emphasized. Additionally, the paper proposes an agentic workflow but does not cite related literature; such citations should be included in the related work section.
- (W2)	Evaluation results: This is the main weakness. The benchmark used in experiments is not publicly available and is poorly described, making it hard to assess the quality of the experiments. The experimental section mentions a "proprietary LLM" without specifying which model is used, adding ambiguity to the evaluation. The explanation quality assessment relies on LLMs, but it is unclear if any manual evaluation by the authors was performed to validate these judgments. The experimental section needs many more results; currently, it's hard for the reviewer to judge the quality of the work.

We note that the authors addressed mostly all the comments from the reviewers.

**Additional Comments On Reviewer Discussion:**

The authors have been proactive in addressing the comments raised by the reviewers, and the reviewers were well engaged responding to the authors.

We agree with the reviewers comments, and recommendations, noting some of the weaknesses that we believe may remain are mentioned in the metareview.

No ethics review raised by the reviewers, and we agree with them.

---

### Decision · Program_Chairs · 2025-01-22

Accept (Poster)